# Temporal Source Recovery for Time-Series Source-Free Unsupervised Domain Adaptation

## Abstract

Source-Free Unsupervised Domain Adaptation (SFUDA) has gained popularity for its ability to adapt pretrained models to target domains without accessing source domains, ensuring source data privacy. While SFUDA is well-developed in visual tasks, its application to Time-Series SFUDA (TS-SFUDA) remains limited due to the challenge of transferring crucial temporal dependencies across domains. Although a few researchers attempt to address this challenge, they rely on specific source pretraining designs, which are impractical as source data owners cannot be expected to follow particular pretraining protocols. To solve this, we propose Temporal Source Recovery (TemSR), a framework that recovers and transfers temporal dependencies for effective TS-SFUDA without requiring source-specific designs. TemSR features a recovery process that employs masking, recovery, and optimization to generate a source-like distribution with recovered source temporal dependencies. To ensure effective recovery, we further design segment-based regularization to restore local dependencies and anchor-based recovery diversity maximization to enhance the diversity of the source-like distribution. With the source-like distribution, the temporal dependencies can be effectively transferred across domains using traditional UDA techniques. Extensive experiments across multiple TS tasks demonstrate the effectiveness of TemSR, even surpassing existing TS-SFUDA method that requires source domain designs.

## 1 Introduction

With the rapid growth of the Internet of Things, Time-Series (TS) data has been increasingly critical in various domains, such as healthcare (Klepl et al., 2024; Jin et al., 2024; Ott et al., 2022) and industrial maintenance (Wang et al., 2024b; Chen et al., 2020). While deep learning models yield promising results in these areas, they heavily depend on extensive labeled data, which is hard to obtain due to high labeling costs. To address this, Unsupervised Domain Adaptation (UDA) methods (Wilson & Cook, 2020; Wang et al., 2024a), which transfer knowledge from a labeled source domain to an unlabeled target domain, have gained attention to reduce label reliance in TS tasks.

Although UDA techniques have proven effective, they typically require access to both source and target domains to bridge domain gaps. However, in many real-world applications, data privacy concerns prevent access to source domain data (Li et al., 2024), leaving only a pretrained model available for adaptation. This challenge significantly limits the applicability of existing UDA methods, as they are not designed for such restricted settings. To address this issue, researchers have recently focused on a more practical scenario, Source-Free Unsupervised Domain Adaptation (SFUDA), which adapts the pretrained model to the target domain without relying on source data, demonstrating promising results. Despite these advancements, most existing techniques were developed for visual tasks and overlook the temporal dependencies inherent in TS data (Ragab et al., 2023b), limiting their generalizability to Time-Series Source-Free Unsupervised Domain Adaptation (TS-SFUDA).

In TS data, temporal dependencies refer to the temporal correlations among time points within a sequence. For effective adaptation, transferring these dependencies from the source to the target domain is essential to learn effective domain-invariant features for TS data (Ragab et al., 2023a; Purushotham et al., 2017). However, without access to source data, directly transferring these dependencies becomes challenging. To address this, recent research (Ragab et al., 2023b) has explored methods to preserve temporal dependencies during source pretraining and restore them during target

adaptation. Although effective, these approaches require specific pretraining designs in the source domain, which are impractical for real-world applications. Thus, a robust TS-SFUDA approach must meet two key criteria: *1. Even without source data, the temporal dependencies can still be transferred across domains; 2. Additional designs during source pretraining should be avoided.*

Following the criteria, we introduce Temporal Source Recovery (TemSR), a novel framework to recover and transfer source temporal dependencies for improved TS-SFUDA. TemSR contains two steps: recovery and enhancement, jointly restoring source temporal dependencies to facilitate transfer using traditional UDA techniques. In the recovery step, we apply masking, recovery, and optimization to generate a source-like distribution with recovered source temporal dependencies. Masked target TS samples are recovered by a recovery model, then optimized to follow a source-like distribution by minimizing their entropy computed using a fixed pretrained source model. With the minimized entropy on source data, the source model can produce deterministic outputs for distributions with source characteristics. By minimizing the entropy of recovered samples, this output constraint can inversely regularize these samples, forcing them to align with the source-like distribution. Meanwhile, this process forces the recovery model to recover the source temporal dependencies required to effectively fill in the masked parts using unmasked time points. However, focusing only on sample-level recovery for long-term patterns may overlook local temporal dependencies, which capture critical short-term trends and are essential for recovering source temporal dependencies. To address this, we improve the optimization as segment-based regularization, enforcing minimal entropy across segments in recovered samples to ensure effective recovery of local dependencies.

A crucial aspect of the recovery process is the masking, which introduces the diversity necessary to effectively recover a source-like distribution. However, this presents challenges: a high masking ratio may lead the recovery model to collapse into constant values for entropy minimization, like zeros, while a low masking ratio may result in insufficient diversity, hindering effective recovery of the source-like distribution. To enhance the recovery, we introduce an anchor-based recovery diversity maximization module, where recovery diversity maximization enhances diversity in recovered samples and anchors ensure this diversity aligns with the source distribution. By effectively enhancing diversity, this module facilitates the recovery of an optimal source-like distribution.

Our contributions are threefold. 1. We design a recovery process involving masking, recovery, and optimization to generate a source-like distribution with recovered source temporal dependencies, which is further refined by segment-based regularization to improve temporal dependency recovery. 2. We design an enhancement module to improve diversity in the source-like distribution through anchor-based recovery diversity maximization, with anchors ensuring this diversity aligns with the source distribution. By effectively enhancing diversity, this module facilitates the recovery of an optimal source-like distribution. 3. Extensive experiments across various TS tasks indicate the effectiveness of TemSR, which even surpasses existing TS-SFUDA method that requires source pretraining designs. Additional analysis on distribution discrepancy changes between source, source-like, and target domains further verify TemSR's ability to recover an effective source-like domain and thus reduce gaps between the source and target domains even without access to the source data.

## 2  RELATED WORK

**Source-Free Unsupervised Domain Adaptation**   To enable effective UDA without access to source data, researchers have explored SFUDA through model- and data-based methods (Fang et al., 2024). Model-based approaches adapt a source pretrained model to the target domain through self-supervised techniques, such as entropy regularization (Mao et al., 2024; Ahmed et al., 2021), pseudo-label generation (Yang et al., 2021; Xie et al., 2022; Ding et al., 2023), and contrastive learning (Zhang et al., 2022; Huang et al., 2021). On the other hand, data-based methods aim to reconstruct the source distribution by selecting relevant data from the target domain (Du et al., 2024; Qiu et al., 2021) or using Generative Adversarial Networks (GANs) to synthesize source-like samples (Kurmi et al., 2021), allowing traditional UDA techniques to be applied. By effectively 'seeing' source distribution in a source-free setting, data-based methods can achieve more stable adaptation by transferring useful information across domains. However, most existing SFUDA algorithms are tailored for visual tasks and overlook crucial temporal dependencies in TS data, limiting their effectiveness in TS-SFUDA. For example, the performance of data-based methods hinges on the quality

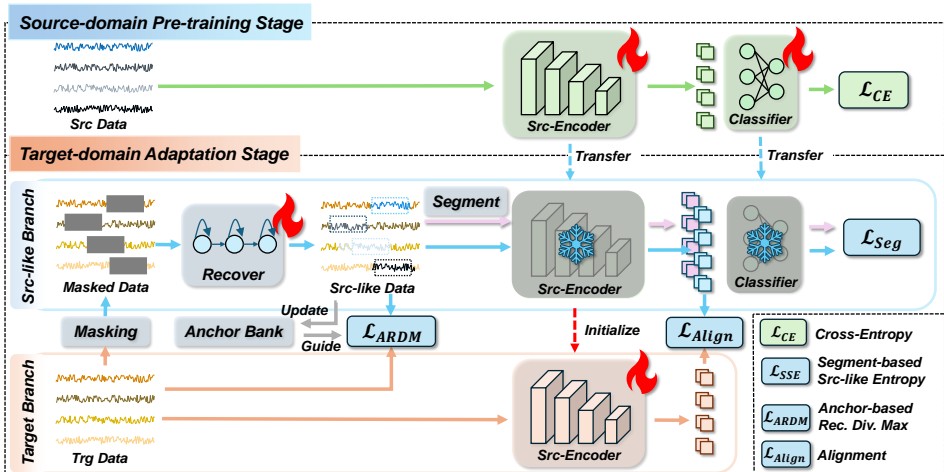

Figure 1: Overall TemSR. An encoder pretrained on the source domain is transferred to the target domain for adaptation without the access to source data, using source-like and target branches. In the source-like branch, masked target samples are recovered. With the fixed source encoder, their entropy is computed via a Segment-based Source-like Entropy loss $\mathcal{L}_{SSE}$ and minimized for optimization to generate a source-like distribution with restored temporal dependencies. Meanwhile, an Anchor-based Recovery Diversity Maximization loss $\mathcal{L}_{ARDM}$ enhances the diversity of the generated distribution for effective recovery. Finally, source-like and target distributions are aligned with an alignment loss $\mathcal{L}_{Align}$, enabling the transfer of temporal dependencies for effective TS-SFUDA.

of generated source distributions. Without considering temporal dependencies, the generated distributions lack key temporal information, significantly hampering adaptation performance in TS tasks.

**Time-Series Unsupervised Domain Adaptation**  To reduce label reliance in TS tasks, UDA methods have been widely applied. The main challenge in TS UDA is transferring temporal dependencies across domains to learn domain-invariant features (Ragab et al., 2023a), typically achieved through metric- and adversarial-based methods. Metric-based methods extract temporal features and align them using statistical measures such as Deep CORAL (Liu & Xue, 2021; He et al., 2023; Cai et al., 2021), while adversarial-based methods leverage discriminators to learn domain-invariant temporal features (Wilson et al., 2020; 2023; Purushotham et al., 2017). To enhance robustness, contrastive learning has been explored to learn discriminative features (Eldele et al., 2023; Ozyurt et al., 2022), and spatial dependencies have also been investigated (Wang et al., 2023; 2024a). Besides TS-related works, video UDA has been explored Sahoo et al. (2021); Wei et al. (2023), which shares similar sequential properties with TS data. However, video UDA methods cannot effectively leverage the unique temporal properties of TS data, limiting their applicability in this area Ozyurt et al. (2022).

Despite their potential, TS UDA methods rely on access to source data, which may not always be feasible due to privacy concerns. This highlights the need for TS-SFUDA, where adaptation is performed without source data. While a few researchers (Ragab et al., 2023b) have explored this, demonstrating the effectiveness of transferring temporal dependencies in TS-SFUDA, they required additional designs in source pretraining to preserve the dependencies. This is impractical, as source data holders cannot be expected to follow specific pretraining steps. To overcome this, we propose TemSR, which effectively transfers temporal dependencies across domains without extra operations during source pretraining, ensuring both practicality and strong performance for TS-SFUDA.

## 3 METHODOLOGY

### 3.1 PROBLEM DEFINITION

Given a labeled source domain $\mathbb{D}_S = \{\boldsymbol{X}_S^i, y_S^i\}_{i=1}^{n_S}$ with $n_S$ samples and an unlabeled target domain $\mathbb{D}_T = \{\boldsymbol{X}_T^i\}_{i=1}^{n_T}$ with $n_T$ samples, $\boldsymbol{X}_S$ and $\boldsymbol{X}_T$ represent TS data with $N$ channels and $L$ time points, and $y_S$ denotes source labels. We aim to train an encoder $\mathcal{F}_\theta$ and a classifier $\mathcal{G}_\phi$ on the source domain, then transfer the pretrained encoder to the target domain without accessing source data.

Given the critical role of temporal dependencies in TS data, transferring these dependencies across domains is key for TS-SFUDA. However, this becomes challenging in the absence of source data. To address this, we propose generating a source-like domain with recovered temporal dependencies, enabling traditional UDA techniques to transfer these dependencies to the target domain.

## 3.2 OVERALL FRAMEWORK

Fig. 1 presents the overall TemSR, where an encoder is pretrained on the source domain and then adapted to the target domain without source data, using both the source-like and target branches. In the source-like branch, target samples are masked and recovered. Using the fixed source encoder, we derive entropy for the recovered samples through segment-based regularization, computing the segment-based source-like entropy loss, which is then minimized for optimization to generate a source-like distribution with restored temporal dependencies. To enhance the diversity of the generated distribution, we introduce an anchor-based recovery diversity maximization loss for better recovery. Finally, the source-like and target distributions are aligned by an alignment loss, effectively transferring temporal dependencies across domains for TS-SFUDA. Further details are provided in following sections, with pesudo-code available in Appendix A.10.

## 3.3 RECOVERY

The recovery process begins with an initialized distribution. Masking introduces diversity into the initialized samples, which are then recovered and optimized to generate a source-like distribution with source temporal dependencies. For more effective temporal recovery, the optimization is further refined as segment-based regularization.

**Initialization**  A critical step in generating an effective source-like distribution is proper initialization, for which we identify two key requirements:

1. The initialized distribution should be close to the source distribution; otherwise, obtaining an effective source-like distribution is difficult.

2. The time points of the initialized samples must be continuous, as random time points would hinder the recovery of source temporal dependencies.

Existing generative methods, such as GANs, fail to meet these requirements (see Appendix A.3), making it difficult to generate an effective source-like distribution with restored temporal dependencies. To solve this, initializing the source distribution using the target distribution offers an effective solution. As UDA typically operates on different but related

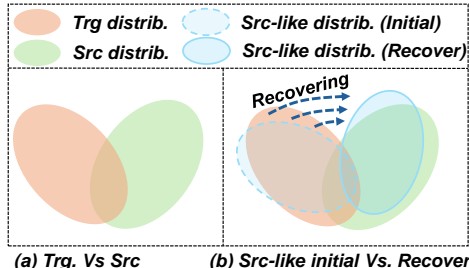

Figure 2: (a) Source and target distributions are distinct but related. (b) Source-like distribution, when initialized from the target distribution, can more easily be optimized to resemble source distribution.

domains, the target distribution is normally not significantly different from the source distribution, as shown in Fig. 2. By initializing a source-like domain with the target domain, we can simplify the optimization process but also preserve the continuity of time points in the samples.

**Masking and Recovery**  With the initialized distribution, we introduce diversity to allow optimization toward the source distribution. Masking is an effective approach, as it not only introduces diversity but also aids in recovering temporal dependencies. By masking portions of TS data, a recovery model is forced to reconstruct masked portions with available information from unmasked parts. To effectively recover the masked data, the model needs to understand how time points are connected and how patterns evolve. This process encourages the model to capture the underlying structure and temporal dependencies in TS data, allowing it to restore these dependencies during recovery. As shown in Fig. 1, portions of the TS sequences are masked, determined by a masking ratio $p_m$ (see sensitivity analysis in Appendix A.8). Given a target sample $X_T^i$, masking generates its masked form $\bar{X}_T^i = M(X_T^i)$, which is recovered by a recovery model $\mathcal{R}_\zeta$ as a source-like sample $X_{Sl}^i = \mathcal{R}_\zeta(M(X_T^i))$. These recovered samples are then optimized to align with the source domain.

**Optimization**  To align the recovered samples with the source domain, we propose leveraging the pretrained source model with entropy minimization as guidance. Entropy minimization is widely used in model adaptation, as models with minimized entropy can produce deterministic outputs, and this ideal output constraint can be inversely employed to guide adaptation (Li et al., 2024; Liang et al., 2020). Inspired by this, we introduce entropy minimization to optimize the recovered samples. With the minimized entropy on source data, the source model can produce deterministic outputs for distributions with source characteristics. By minimizing the entropy computed by the fixed source model for recovered samples, this constraint can inversely regularize the samples, forcing them to align with the source distribution. Here, the recovery model is forced to capture source temporal dependencies, as only by understanding these dependencies can the model effectively reconstruct masked parts, minimize entropy, and ensure recovered samples align with the source distribution.

While the recovery process can generate source-like distributions with recovered temporal dependencies, it primarily focuses on sample-level recovery for long-term patterns, overlooking local temporal dependencies. These local dependencies offer short-term context, enabling the model to infer with local information that may not be apparent in broader trends. This highlights the importance of recovering local dependencies to restore natural temporal patterns and enhance overall temporal recovery. Thus, we improve the optimization as segment-based regularization, further optimizing segments that capture local dependencies to have minimized entropy, aligning them with source distributions. Three types of segments are extracted from the recovered sample $X_{Sl}^i$ with an extraction proportion $p_s$, capturing local information from different regions (see examples in Appendix A.7):

1. Early Segment $X_{Sl,E}^i$: Extracts the first $p_s$ proportion of the sequence, capturing local information at the early stage of the recovered sample $X_{Sl}^i$.

2. Late Segment $X_{Sl,L}^i$: Extracts the last $p_s$ proportion capturing local information at the later stage.

3. Segment with Recovered Parts $X_{Sl,R}^i$: Extracts all recovered portions to ensure they have minimized entropy and align with the source-like distribution.

These segments effectively capture local temporal dependencies. Along with the complete recovered sample $X_{Sl}^i$ for sample-level recovery, denoted as $X_{Sl,C}^i$ for consistency, we minimize their entropy:

$$\mathcal{L}_{SegEnt} = \sum_{k \in \{C,E,L,R\}} -\sum_i \mathcal{G}_\phi(\mathcal{F}_\theta(X_{Sl,k}^i)) \log \mathcal{G}_\phi(\mathcal{F}_\theta(X_{Sl,k}^i)). \tag{1}$$

Besides minimizing the entropy of these segments, ensuring similar entropy across segments is also crucial. Large differences in entropy between segments may indicate disruptions in the flow of temporal information, suggesting the model has failed to capture smooth dependencies in recovered TS sequences. To address this, the recovered samples are designed to retain consistent entropy values across these segments, as shown in Eq. (2). By enforcing similar entropy across different segments, TemSR maintains a uniform level of temporal structure.

$$\mathcal{L}_{SegSim} = \sum_{(k,s) \in \{C,E,L,R\}} \left( \sum_i \mathcal{G}_\phi(\mathcal{F}_\theta(X_{Sl,k}^i)) \log \mathcal{G}_\phi(\mathcal{F}_\theta(X_{Sl,k}^i)) \right.$$

$$\left. -\sum_i \mathcal{G}_\phi(\mathcal{F}_\theta(X_{Sl,s}^i)) \log \mathcal{G}_\phi(\mathcal{F}_\theta(X_{Sl,s}^i)) \right). \tag{2}$$

By combining the two losses, we define the segment-based source-like entropy loss as $\mathcal{L}_{Seg} = \mathcal{L}_{SegEnt} + \mathcal{L}_{SimEnt}$. By minimizing $\mathcal{L}_{Seg}$, we effectively generate a source-like distribution with recovered source temporal dependencies.

## 3.4 ENHANCEMENT

To optimize the initial distribution as a source-like distribution, masking introduces the essential diversity required for effective recovery. However, masking presents challenges. A large masking ratio can introduce sufficient diversity, increasing the chances of finding an optimal solution. However, it risks model collapse, where the recovery model shortcuts the learning process by filling masked parts with constant values, minimizing entropy without capturing the true underlying structure, as proof in Appendix A.1. On the other hand, using a small masking ratio avoids this collapse but fails to provide enough diversity for the model to learn an optimal source-like distribution.

**Anchor-based Recovery Diversity Maximization**   To effectively enhance diversity for optimal recovery, we introduce the anchor-based recovery diversity maximization module. This module encourages recovery diversity by maximizing the distance between recovered samples and their original samples. By pushing the recovered samples to diverge from their original forms, the samples are forced to enhance diversity (see proof in Appendix A.2), allowing to explore a broader range of features that are crucial for capturing the complexity of the source distribution. However, without proper constraints, this recovery diversity maximization may cause the recovered samples to deviate in unintended directions, as shown in Fig. 3 (a), leading to distributions that are not aligned with the source domain and hurting performance. To prevent this, we further introduce anchors to guide the process and ensure that the diversity remain consistent with the source distribution. Anchors act as reference points as shown in Fig. 3 (b), balancing diversity with fidelity to the source domain.

**Anchor Generation with Anchor Bank**   To effectively guide optimization toward the source distribution, generating high-quality anchors is crucial, as poor anchors can mislead the model and degrade performance. For optimal guidance, these anchors must closely align with the source distribution. Thus, we propose selecting recovered samples with the lowest entropy, as they are more likely to reflect the source distribution and serve as ideal guides for the recovery process. While a simple approach is to select low-entropy samples from each batch, this may miss optimal candidates due to batch randomness. To address this, we implement an anchor bank, inspired by Wu et al. (2018), to store all recovered samples with their entropy: $\mathbb{A} = \{\boldsymbol{X}_{Sl}^i, H(\boldsymbol{X}_{Sl}^i)\}_{i=1}^{n_T}$, where

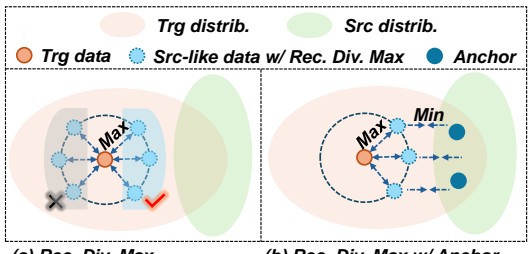

Figure 3: (a) Recovery diversity maximization may cause the recovered samples to deviate in unintended directions without proper constraints. (b) Anchors act as reference points, balancing diversity with fidelity to the source domain.

$H(\boldsymbol{X}_{Sl}^i)$ is the entropy computed by the source model. To ensure its quality, the anchor bank is continuously updated during adaptation, as shown in Fig. 1. From the anchor bank, we extract the top $k$ samples with the lowest entropy, denoted by $\mathbb{A}_k = \{\boldsymbol{X}_A^j\}_j^k$, and compute a representative anchor by averaging these samples: $\bar{\boldsymbol{X}}_A = \sum_j^k \boldsymbol{X}_A^j/k$. The value of $k$ is set by an anchor ratio, allowing adjustment based on dataset sizes. Further analysis of the anchor ratio is provided in Appendix A.8.

**Objectives**   We have two key objectives: 1. Recovery Diversity Maximization: Maximize the distances between the recovered samples and their original samples; 2. Anchor Guidance: Minimize the distances between the recovered samples and the anchor sample. However, directly pushing all recovered samples toward the anchor risks collapse, where diversity is lost as all samples converge to a single point. To prevent this, we introduce an additional objective that maximizes the distances between any two recovered samples, ensuring variations among them. To achieve these objectives, the InfoNCE loss for contrastive learning is adopted (Eldele et al., 2021), which pulls the recovered samples toward the anchor while pushing them apart from each other and their original forms. Particularly, given recovered source-like samples $\boldsymbol{X}_{Sl}^i$, original target samples $\boldsymbol{X}_T^i$, and the anchor $\bar{\boldsymbol{X}}_A$, the anchor-based recovery diversity maximization loss is defined as Eq. (3), where $B$ is batch size, $\mathcal{S}(\boldsymbol{i},\boldsymbol{j}) = \exp\left(m(\boldsymbol{i},\boldsymbol{j})/\tau\right)$, with $m(\boldsymbol{i},\boldsymbol{j}) = \mathcal{F}_\theta(\boldsymbol{i})(\mathcal{F}_\theta(\boldsymbol{j}))^T$ measuring the difference of samples.

$$\mathcal{L}_{ARDM} = -\frac{1}{B}\sum_{i=1}^B \log \frac{\mathcal{S}(\boldsymbol{X}_{Sl}^i, \bar{\boldsymbol{X}}_A)}{\mathcal{S}(\boldsymbol{X}_{Sl}^i, \bar{\boldsymbol{X}}_A) + \mathcal{S}(\boldsymbol{X}_{Sl}^i, \boldsymbol{X}_T^i) + \sum_{k \neq i} \mathcal{S}(\boldsymbol{X}_{Sl}^i, \boldsymbol{X}_{Sl}^k)}. \tag{3}$$

### 3.5 ADAPTATION

Once the source-like distribution with source temporal dependencies is generated, we transfer this information to the target domain for adaptation. With the source temporal dependencies already recovered, traditional UDA techniques, such as metric-based or adversarial-based methods, can be effectively utilized for this transfer. For adaptation, we fine-tune the target encoder $\bar{\mathcal{F}}_{\bar{\theta}}$, initialized from the pretrained source encoder $\mathcal{F}_\theta$, to adapt to the target domain. To further preserve target

domain information, we incorporate target entropy minimization following Liang et al. (2020), i.e., $\mathcal{L}_{TrgEnt} = -\sum_i \mathcal{G}_\phi(\bar{\mathcal{F}}_{\bar{\theta}}(\boldsymbol{X}_T^i)) \log \mathcal{G}_\phi(\bar{\mathcal{F}}_{\bar{\theta}}(\boldsymbol{X}_T^i))$. The final loss function is shown in Eq. (4), including the alignment loss $\mathcal{L}_{Align}$ computed by Deep CORAL (Sun et al., 2017; Wang et al., 2024a).

$$\min \mathcal{L} = \lambda_{Seg}\mathcal{L}_{Seg} + \lambda_{ARDM}\mathcal{L}_{ARDM} + \mathcal{L}_{Align} + \mathcal{L}_{TrgEnt}. \tag{4}$$

Notably, the source-like distribution may have poor quality during initial epochs, and adaptation at this stage could cause negative transfer. To solve this, we divide the adaptation process into source-like optimization and transfer phases. First, the source-like distribution is optimized over several epochs to enhance its quality. This enhanced source-like distribution is then used to transfer dependencies to the target encoder during the transfer phase for effective domain adaptation.

## 4 EXPERIMENTS

### 4.1 DATASETS AND SETTINGS

**Datasets** To comprehensively evaluate TemSR, we selected three crucial TS tasks: Human Activity Recognition (HAR) on the UCI-HAR dataset (Anguita et al., 2013), Sleep Stage Classification (SSC) on the Sleep-EDF dataset (Goldberger et al., 2000), and Machine Fault Diagnosis (MFD) (Lessmeier et al., 2016). Each task is assessed through ten cross-domain scenarios by following Ragab et al. (2023a). Detailed descriptions and preprocessing are provided in Appendix A.4.

**Unified Training Scheme** To ensure fair comparisons with SOTAs, we utilized a consistent three-layer CNN backbone and adhered to identical training configurations as Ragab et al. (2023b). To consider potential data imbalances and provide comprehensive evaluations, we used the Macro F1-score (MF1) as the primary metric. The mean and standard deviation of MF1 are reported across three runs for each cross-domain scenario. Full details are available in Appendix A.5.

### 4.2 COMPARISONS WITH STATE-OF-THE-ARTS

Table 1: Detailed results of the ten HAR cross-domain scenarios in terms of MF1 score (%).

| Models | SF | 2→11 | 12→16 | 9→18 | 6→23 | 7→13 | 18→27 | 20→5 | 24→8 | 28→27 | 30→20 | AVG |
|---|---|---|---|---|---|---|---|---|---|---|---|---|
| SRC | † | 95.69±5.72 | 67.13±9.83 | 70.07±4.71 | 81.01±14.9 | 84.5±12.08 | 85.95±5.00 | 63.30±4.13 | 71.59±8.56 | 50.24±5.92 | 67.91±9.21 | 73.73±2.68 |
| TRG | † | 100.0±0.00 | 98.50±1.30 | 100.0±0.00 | 100.0±0.00 | 100.0±0.00 | 100.0±0.00 | 97.21±3.08 | 100.0±0.00 | 100.0±0.00 | 88.61±9.36 | 98.43±2.84 |
| DANN | ✗ | 98.09±1.68 | 62.08±1.69 | 70.7±11.36 | 85.6±15.71 | 93.33±0.00 | **100.0±0.00** | 78.41±7.67 | 87.99±9.41 | 97.47±1.00 | **87.25±0.81** | 86.09±4.86 |
| CDAN | ✗ | 98.19±1.57 | 61.20±3.27 | 71.3±14.64 | 96.73±0.00 | 93.33±0.00 | 99.61±0.67 | 82.02±5.43 | 98.59±2.44 | 99.12±1.52 | 80.70±7.43 | 88.07±1.22 |
| CoDATs | ✗ | 86.65±4.28 | 61.03±2.33 | 80.51±8.47 | 92.08±4.39 | 92.61±0.51 | 97.67±1.02 | 82.81±7.05 | 94.69±1.81 | 92.29±9.25 | 80.44±5.04 | 86.07±2.88 |
| CLUDA | ✗ | 80.33±3.81 | 66.67±2.24 | 70.35±2.13 | 91.14±1.70 | 95.28±2.62 | 100.0±0.00 | 80.73±3.24 | 91.67±3.15 | 98.96±1.47 | 80.43±2.34 | 85.55±1.24 |
| RAINCOAT | ✗ | **100.0±0.00** | 76.28±3.18 | 77.35±3.70 | 98.14±1.20 | **100.0±0.00** | **100.0±0.00** | **85.73±3.02** | 97.67±2.31 | **100.0±0.00** | 86.46±1.04 | **92.16±0.83** |
| SHOT | ✓ | **100.0±0.00** | 70.76±6.22 | 70.19±8.99 | 98.91±1.89 | 93.01±0.57 | 92.93±2.79 | 69.66±1.06 | 88.58±3.94 | 90.39±3.11 | 75.47±1.96 | 84.99±2.00 |
| NRC | ✓ | 97.02±2.82 | **72.18±0.59** | 63.10±4.84 | 96.41±1.33 | 89.13±0.54 | 100.0±0.00 | 81.82±1.19 | 92.97±3.21 | 98.43±0.88 | 82.97±2.71 | 87.40±0.34 |
| AaD | ✓ | 98.51±2.58 | 66.15±6.15 | 68.33±11.9 | 98.07±1.71 | 89.41±2.86 | 100.0±0.00 | 80.75±2.72 | 94.69±3.42 | 84.85±13.1 | 77.77±1.43 | 85.85±1.29 |
| BAIT | ✓ | 98.88±1.93 | 56.65±2.54 | 80.4±13.43 | 100.0±0.00 | 97.43±3.59 | 100.0±0.00 | 80.91±1.60 | 100.0±0.00 | 98.66±1.30 |  | 89.69±1.23 |
| MAPU | ✓ | **100.0±0.00** | 67.96±4.62 | 82.77±2.54 | 97.82±1.89 | 99.29±1.22 | 100.0±0.00 | 82.88±3.68 | 96.48±3.09 | 96.01±3.19 | 85.43±3.84 | 90.86±0.98 |
| TemSR | ✓ | **100.0±0.00** | 64.21±3.04 | **93.65±2.02** | 97.82±1.89 | 98.95±0.01 | **100.0±0.00** | 82.32±0.73 | **100.0±0.00** | **100.0±0.00** | 84.10±5.52 | 92.10±0.33 |

Table 2: Detailed results of the ten SSC cross-domain scenarios in terms of MF1 score (%).

| Models | SF | 16→1 | 9→14 | 12→5 | 7→18 | 0→11 | 3→19 | 18→12 | 13→17 | 5→15 | 6→2 | AVG |
|---|---|---|---|---|---|---|---|---|---|---|---|---|
| SRC | † | 52.93±3.42 | 63.99±8.04 | 48.79±3.31 | 62.33±3.86 | 50.43±6.26 | 47.38±3.36 | 38.35±2.03 | 43.80±0.12 | 60.13±6.36 | 55.67±2.20 | 52.38±0.47 |
| TRG | † | 81.52±2.06 | 75.79±0.88 | 73.87±1.43 | 77.74±1.86 | 68.26±0.73 | 78.79±1.49 | 73.51±1.73 | 70.39±0.75 | 72.17±1.99 | 82.11±1.13 | 75.41±0.43 |
| DANN | ✗ | 58.68±3.29 | 64.29±1.08 | 64.65±1.83 | 69.54±3.00 | 44.13±5.84 | 64.09±4.48 | 54.33±4.81 | 52.31±1.70 | 68.03±0.29 | 71.78±2.24 | 61.18±2.31 |
| CDAN | ✗ | 59.65±4.96 | 64.18±6.37 | 64.43±1.17 | 67.61±3.55 | 39.38±3.28 | 60.19±1.16 | 40.46±6.79 | 40.82±8.87 | 65.22±6.73 | 68.81±1.86 | 57.07±1.79 |
| CoDATs | ✗ | 63.84±3.36 | 63.51±6.92 | 52.54±5.94 | 66.06±2.48 | 46.28±5.99 | 66.15±4.46 | 47.84±5.59 | 38.17±10.8 | 72.62±3.07 | 61.59±13.1 | 57.86±0.76 |
| CLUDA | ✗ | 55.67±1.21 | 64.33±1.24 | 60.12±4.55 | 64.35±1.55 | 46.35±2.22 | 45.56±1.34 | 51.12±6.77 | 64.55±1.21 | 61.12±3.34 |  | 57.79±1.37 |
| RAINCOAT | ✗ | 59.04±2.02 | 68.04±1.18 | 62.20±3.22 | 66.77±1.56 | 49.17±2.70 | **68.89±0.66** | 49.40±1.25 | 50.71±6.68 | **73.53±0.51** | 72.09±2.38 | **61.98±1.48** |
| SHOT | ✓ | 59.07±2.14 | 69.93±0.46 | 62.11±1.62 | 69.74±1.22 | **50.78±1.90** | 65.44±1.06 | 48.14±11.2 | 56.41±1.60 | 55.51±9.37 | 64.56±2.16 | 60.16±3.82 |
| NRC | ✓ | 52.09±1.89 | 58.52±0.66 | 59.87±2.48 | 66.18±0.25 | 47.55±1.72 | 64.65±2.25 | 52.86±6.60 | 56.93±2.89 | 61.89±5.94 | 66.54±2.29 | 58.70±2.79 |
| AaD | ✓ | 57.04±2.03 | 65.27±1.69 | 61.84±1.74 | 67.35±1.48 | 44.04±2.18 | 52.42±4.55 | 40.86±8.43 | **58.28±6.97** | 63.06±12.3 | 59.29±2.90 | 56.94±3.52 |
| BAIT | ✓ | 56.83±1.17 | 71.84±1.18 | 65.57±2.15 | 71.12±1.45 | 42.30±2.61 | 59.56±1.87 | 53.53±1.89 | 53.03±3.53 | 60.53±5.08 | 63.69±1.04 | 59.80±0.60 |
| MAPU | ✓ | **63.85±4.63** | **74.73±0.64** | 64.08±2.21 | **74.21±0.58** | 43.36±5.49 | 59.03±3.60 | 52.82±4.94 | 48.09±2.25 | 67.04±1.22 | 58.98±1.07 | 60.61±1.28 |
| **TemSR** | ✓ | 62.51±1.09 | 72.60±0.74 | **66.70±1.91** | 72.15±1.01 | 49.62±1.88 | 65.87±0.53 | **60.32±0.97** | 57.56±2.07 | 66.50±2.07 | 64.82±1.78 | **63.86±0.58** |

For comparisons, we evaluated both conventional UDA methods and SFUDA techniques by following Ragab et al. (2023b); Yang et al. (2021; 2022). Conventional UDA methods include DANN (Ganin et al., 2016), CDAN (Long et al., 2018), CoDATS (Wilson et al., 2020), CLUDA Ozyurt et al. (2022), and RAINCOAT He et al. (2023), while SFUDA methods include SHOT (Liang et al.,

Table 3: Detailed results of the ten MFD cross-domain scenarios in terms of MF1 score (%).

| Models | SF | 0→1 | 1→0 | 1→2 | 2→3 | 3→1 | 0→3 | 1→3 | 2→1 | 3→0 | 3→2 | AVG |
|---|---|---|---|---|---|---|---|---|---|---|---|---|
| SRC | † | 26.26±5.04 | 68.63±6.22 | 72.66±0.95 | 96.90±1.38 | 99.02±1.07 | 42.13±8.06 | 96.25±3.72 | 86.96±0.58 | 46.42±2.42 | 71.71±6.54 | 70.69±2.61 |
| TRG | † | 100.0±0.00 | 97.88±1.60 | 99.92±0.14 | 100.0±0.00 | 100.0±0.00 | 100.0±0.00 | 100.0±0.00 | 100.0±0.00 | 97.88±1.60 | 99.92±0.14 | 99.56±2.31 |
| DANN | ✗ | 83.44±1.72 | 51.52±0.38 | 84.19±2.10 | **99.95±0.09** | **100.0±0.00** | 77.65±9.41 | **99.97±0.04** | **99.75±0.14** | 50.85±1.74 | 72.32±22.3 | 81.96±2.89 |
| CDAN | ✗ | 84.97±0.62 | 52.39±0.49 | 85.96±0.90 | 99.70±0.45 | **100.0±0.00** | 85.38±0.42 | **100.0±0.00** | 99.02±0.90 | 62.17±6.32 | 79.76±2.75 | 84.93±1.47 |
| CoDATs | ✗ | 67.42±13.3 | 49.92±13.7 | **89.05±4.73** | 99.21±0.79 | 99.92±0.14 | 55.68±3.07 | 99.95±0.09 | **99.75±0.29** | 51.77±1.86 | 83.36±1.25 | 79.60±1.27 |
| CLUDA | ✗ | 84.43±1.43 | 55.66±5.76 | 81.12±1.20 | 91.13±1.32 | 93.44±1.26 | 89.94±2.33 | 97.12±0.98 | 91.23±0.88 | 73.35±3.44 | 79.98±6.67 | 83.74±1.32 |
| RAINCOAT | ✗ | 88.09±1.40 | 59.41±6.61 | 83.87±0.69 | 93.67±1.15 | 94.95±0.71 | 91.19±0.95 | 97.73±0.84 | 92.53±0.79 | 78.45±2.84 | **84.61±0.95** | 86.45±1.12 |
| SHOT | ✓ | 41.99±2.78 | 57.00±0.09 | 80.70±1.49 | 99.48±0.31 | 99.95±0.05 | 83.63±2.32 | 89.33±3.50 | 88.98±1.59 | 72.89±7.84 | 71.38±2.31 | 78.53±1.98 |
| NRC | ✓ | 73.99±1.36 | 74.88±8.81 | 69.23±0.75 | 78.04±11.3 | 71.48±4.59 | 70.88±1.75 | 70.35±6.80 | 72.10±1.34 | 63.67±5.57 | 61.52±3.20 | 70.61±1.60 |
| AaD | ✓ | 71.72±3.96 | 75.33±4.65 | 78.31±2.26 | 90.07±7.02 | 87.45±11.7 | 89.35±2.22 | **100.0±0.00** | 96.49±3.04 | 72.42±4.47 | 74.56±6.80 | 83.57±2.46 |
| BAIT | ✓ | 83.1±14.69 | 60.51±6.43 | 75.9±12.51 | 95.57±2.85 | **100.0±0.00** | 82.12±15.5 | **100.0±0.00** | 85.12±1.49 | 67.21±3.33 | 83.37±6.34 | 83.29±4.60 |
| MAPU | ✓ | 99.43±0.51 | 77.42±0.16 | 85.78±7.38 | 99.67±0.50 | 99.97±0.05 | 85.63±2.44 | **100.0±0.00** | 94.38±0.62 | **88.47±1.99** | 81.51±2.43 | 91.22±1.08 |
| **TemSR** | ✓ | **99.97±0.05** | **87.03±4.05** | 84.47±5.88 | 95.23±3.85 | **100.0±0.00** | **99.95±0.05** | **100.0±0.00** | 96.67±4.21 | 87.17±1.56 | 81.96±5.09 | **93.24±1.83** |

2020), NRC (Yang et al., 2021), AaD (Yang et al., 2022), BAIT (Yang et al., 2023), and MAPU (Ragab et al., 2023b). These baselines are introduced in Appendix A.6. Additionally, we report results for source (SRC)-only and target (TRG)-only models to provide the lower and upper bounds of adaptation. For clarity, lower/upper bounds are denoted by †, conventional UDA methods by ✗, and SFUDA methods by ✓. We adopted all baseline results, except BAIT, from Ragab et al. (2023b), where each method used the same backbone as ours for fairness. BAIT, a visual-based method for generating source-like distributions, was implemented with the same backbone and its publicly available code. Among the SFUDA methods, only MAPU is designed for TS tasks to transfer temporal dependencies, though it requires additional pretraining designs in source domain.

The comparisons for HAR, SSC, and MFD datasets are presented in Tables 1, 2, and 3, respectively. The results show that although RAINCOAT outperforms our method on HAR, it is a traditional UDA method that requires access to the source domain during adaptation. In contrast, our method operates without source data and still achieves comparable performance, highlighting its effectiveness. Among SFUDA methods, the methods considering temporal dependencies, including MAPU and our approach, generally outperform other SFUDA in most cross-domain scenarios. Regarding average performance, MAPU and our method achieve the second-best and best results, respectively, demonstrating the importance of capturing temporal dependencies in TS-SFUDA. Specifically, when compared to the best methods that do not consider temporal dependencies (i.e., BAIT, SHOT, and AaD on the respective datasets), our approach yields significant improvements of 2.41%, 3.70%, and 9.67% on the three datasets. Even compared with MAPU, our method still improves by 1.24%, 3.25%, and 2.02%. Notably, MAPU relies on source pretraining designs to capture temporal dependencies, limiting its practicality. In contrast, our approach adapts entirely in the target domain without any source pretraining operations. Moreover, TemSR effectively recovers the source distribution during adaptation, facilitating a more effective transfer of temporal dependencies and thereby achieving improved and robust performance. These results underscore that without relying on source pretraining designs, TemSR can still transfer temporal dependencies to achieve SOTA performance in TS-SFUDA, even surpassing the existing method that depends on such designs.

## 4.3 Ablation Study

To validate the effectiveness of key modules, e.g., $\mathcal{L}_{Seg}$ and $\mathcal{L}_{ARDM}$, for recovering a source-like distribution, we conducted the ablation study using four types of variants. The first variant, 'Src-like only', uses the source-like branch directly for target prediction. The source-like branch is designed to generate source distributions with recovered temporal dependencies, so we test whether leveraging it for prediction, rather than adaptation, is a feasible approach. Second, we tested variants for the components of $\mathcal{L}_{Seg}$. The 'w/o $\mathcal{L}_{Seg}$' variant removes segment-based regularization, replacing it with sample-level entropy minimization for source-like samples, to evaluate the importance of recovering local temporal dependencies for effective temporal recovery. Variants 'w/o Early', 'w/o Late', 'w/o Recover', and 'w/o Complete' involve the removal of specific segments to determine their individual contributions. 'w/o $\mathcal{L}_{SegSim}$' excludes the segment similarity to assess the necessity of ensuring smooth dependencies across segments. Third, we tested variants for the components of $\mathcal{L}_{ARDM}$. The variant 'w/o $\mathcal{L}_{ARDM}$' removes the whole loss, aiming to evaluate whether the diversity facilitated by this module is necessary for optimal performance. The 'w/o Anchor' variant removes both the anchors and their associated objectives, testing the overall utility of anchors. 'w/o Add. Obj.' excludes only the additional objective while retaining the anchors. The 'w/o Anchor

Bank' variant removes the anchor bank and instead generates anchors within each batch, testing whether the anchor bank is essential for producing the high-quality anchor. The final variant, 'w/ Random Init.' randomly initializes the source-like domain, testing the effectiveness of initializing this domain using the target domain.

The results in Table 4 summarize the average performance across all cross-domain cases, with detailed results provided in Appendix A.8. Here, four key insights emerge. First, the 'Src-like only' variant performs poorly. While the recovered samples successfully align with a source-like distribution, the masking process distorts their original samples, causing them to lose sample-specific information, so directly using these recovered samples for prediction significantly weakens performance. This demonstrates that it is more effective to use the source-like distribution for transferring knowledge to the target encoder rather than for prediction.

Table 4: Ablation study for HAR, SSC, and MFD (%).

| Variants | HAR | SSC | MFD |
|---|---|---|---|
| Src-like Only | 17.68±7.89 | 13.44±2.07 | 19.29±4.66 |
| w/o $\mathcal{L}_{Seg}$ | 90.72±1.27 | 62.74±0.81 | 92.33±2.15 |
| w/o Early | 91.20±0.81 | 62.97±1.30 | 92.17±2.47 |
| w/o Late | 91.16±1.35 | 62.95±1.27 | 92.95±3.19 |
| w/o Recover | 91.93±0.95 | 63.46±0.37 | 92.96±0.15 |
| w/o Complete | 91.04±0.72 | 62.77±1.32 | 92.27±3.35 |
| w/o $\mathcal{L}_{SegSim}$ | 91.50±0.95 | 63.49±0.56 | 92.94±2.67 |
| w/o $\mathcal{L}_{ARDM}$ | 90.00±2.74 | 62.84±1.44 | 92.46±2.41 |
| w/o Anchor | 88.91±1.76 | 62.59±0.97 | 91.49±2.34 |
| w/o Add. Obj. | 90.13±1.68 | 63.43±0.50 | 92.79±3.24 |
| w/o Anchor Bank | 91.97±0.97 | 63.23±0.11 | 93.09±2.32 |
| w/ Random Init. | 91.97±0.92 | 63.36±0.20 | 91.72±2.54 |
| TemSR | **92.10±0.33** | **63.86±0.58** | **93.24±1.83** |

Second, we can observe the effectiveness of the components in $\mathcal{L}_{Seg}$. Removing $\mathcal{L}_{Seg}$ causes significant performance drops, highlighting the importance of recovering local temporal dependencies. Among its components, the 'Complete' is the most significant, as it captures global dependencies. When this component is removed, only local dependencies are captured, which adversely affects the model's performance. Further, the early and late segments are relatively more impactful than the recovered segment. This is likely because the early and late segments sometimes intersect with the recovered parts. However, this does not diminish the importance of the recovered segment, as it focuses on the masked parts, encouraging them to align with source temporal dependencies and further enhancing performance. Additionally, removing $\mathcal{L}_{SegSim}$ causes a notable decline, confirming its effectiveness.

Third, $\mathcal{L}_{ARDM}$ is critical for maintaining diversity among recovered samples. Removing this loss leads to significant performance degradation, as the lack of diversity hinders the generation of an optimal source-like distribution. Meanwhile, removing anchors also causes notable drops, especially under small masking ratios, due to insufficient diversity among the recovered samples. While using anchors without the additional objective improves performance, it risks convergence to a collapsed solution, showing the necessity of the additional objective. Similarly, removing the anchor bank results in lower-quality anchors when generated per batch, reducing adaptation effectiveness. Finally, random initialization of the source-like domain severely reduces performance and increases standard deviation, highlighting the difficulty in identifying an optimal solution without leveraging the target domain for initialization.

## 4.4 SENSITIVITY ANALYSIS

We conducted sensitivity analysis for TemSR, focusing on key hyperparameters: $\lambda_{Seg}$ and $\lambda_{ARDM}$, which control the effects of the losses $\mathcal{L}_{Seg}$ and $\mathcal{L}_{ARDM}$. We adopted a wide range—[1e-3, 1e-2, 1e-1, 1, 10, 50, 100]—to assess TemSR's sensitivity to these large variations, with larger values indicating greater impacts.

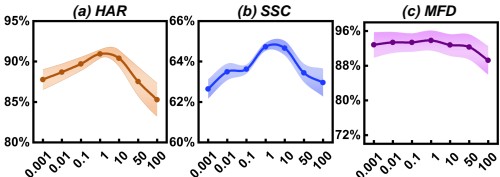

Figure 4: Analysis for $\lambda_{Seg}$.

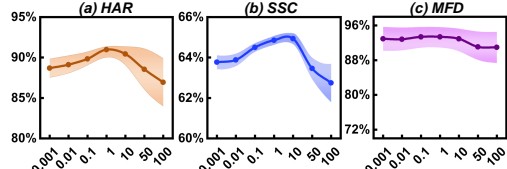

Figure 5: Analysis for $\lambda_{ARDM}$.

Fig. 4 and 5 present the analysis for $\lambda_{Seg}$ and $\lambda_{ARDM}$, respectively. The results show that the performance of TemSR improves as $\lambda_{Seg}$ and $\lambda_{ARDM}$ increase, indicating that greater weights on these losses enhance performance, further highlighting their effectiveness. However, performance drops sharply when these values become too large, e.g., 50 or 100. For instance, with $\lambda_{Seg} = 10 \rightarrow$ 100, the performance on HAR decreases significantly, i.e., from around 91% to 85%. A similar trend is observed with $\lambda_{ARDM}$. These drops occur because, at higher values, the individual loss term dominates the adaptation process, overshadowing the contributions of other losses and thus negatively impacting adaptation. Meanwhile, excessive values also lead to instability, especially at 100. Based on these findings, the optimal range for both $\lambda_{Seg}$ and $\lambda_{ARDM}$ is between 1 and 10, offering a broad range to easily facilitate optimal performance for TemSR.

### 4.5 Distribution Discrepancy Changes

The core objective of TemSR is to recover a source-like domain and then perform domain adaptation. This requires ensuring that the recovered source-like distribution closely resembles the source distribution and that the domain discrepancy between the source-like and target domains is minimized. By achieving so, this process can effectively reduce the gap between the source and target domains. To present this intuitively, we visualized the evolution of distribution discrepancies between source (SRC)-like, source (SRC), and target (TRG) domains, during the adaptation stage. The visualization

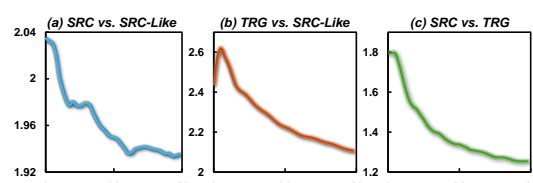

Figure 6: Distribution discrepancies changes (Source domain used only for computing discrepancy without directly involved in adaptation).

is shown in Fig. 6, where discrepancies are quantified using the KL divergence, a standard metric for comparing distributions (Zhang et al., 2024). Notably, in this visualization, the source distribution is used only for calculating discrepancies and is not directly involved in the adaptation process.

From the figure, we observe that the discrepancy between the source and source-like domains decreases steadily during the adaptation stage, indicating that the recovered source-like distribution increasingly resembles the source distribution. Meanwhile, during the initial epochs without alignment, we also notice an increase in the domain gap between the target and source-like domains. After these early stages and the alignment begins, the domain gap between the target and source-like domains gradually diminishes. By the end of adaptation, the overall domain discrepancy between the source and target domains is effectively reduced, demonstrating the capability of TemSR to align the two domains without requiring direct access to the source data.

## 5 Conclusion

To transfer temporal dependencies across domains for effective TS-SFUDA without relying on specific source pretraining designs, we propose the Temporal Source Recovery (TemSR) framework. TemSR aims to recover and transfer source temporal dependencies by generating a source-like time-series distribution. The framework features a recovery process that employs masking, recovery, and optimization to create the source-like distribution with recovered temporal dependencies. For effective recovery, we further improve the optimization as segment-based regularization to restore local temporal dependencies and design an anchor-based recovery diversity maximization loss to enhance diversity in the source-like distribution. The recovered source-like distribution is then adapted to the target domain using traditional UDA techniques. Additional analysis of distribution discrepancy changes between source, source-like, and target domains confirms TemSR's ability to recover and align the source-like domain, ultimately reducing gaps between the source and target domains. Extensive experiments further demonstrate the effectiveness of TemSR, achieving SOTA performance and even surpassing the existing TS-SFUDA method that relies on source-specific designs.

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

## A APPENDIX

### A.1 TRIVIAL SOLUTIONS WITH LARGE MASKING RATIO

**Theorem 1** *With a high masking ratio, the recovery model is prone to collapsing to a constant value for the source-like domain, thus impairing the performance of domain adaptation.*

**Proof:**

**Given Conditions**

- $X_T^i$ is a time-series sample from the target domain;
- $M(X_T^i)$ is the masking operation applied to $X_T^i$, with a masking ratio $p_m$, where $p_m$ represents the proportion of the input that is masked.;
- $\mathcal{R}_\zeta$ is the recovery model, parameterized by $\zeta$, which recovers a source-like sample $X_{Sl}^i = \mathcal{R}_\zeta(M(X_T^i))$ from the masked input;
- $\mathcal{F}_\theta$ is the fixed pretrained encoder for the source-like branch, aiming to extract features $\mathbf{z}$ from the recovered sample $X_{Sl}^i$;
- $p(\mathbf{z})$ denotes the probability distribution of the feature representations.

The entropy of the feature distribution is given by the following, and the training objective is minimizing this entropy,

$$H(p(\mathbf{z})) = - \int p(\mathbf{z}) \log p(\mathbf{z}) \, d\mathbf{z}. \tag{5}$$

**Feature Collapse in High Masking Ratio** As the masking ratio $p_m$ increases toward 1, the masked sample $M(X_T^i)$ contains minimal information about the original target data $X_T^i$. Consequently, the recovery model $\mathcal{R}_\zeta$ faces increasing difficulty in reconstructing meaningful samples. To achieve the training objectives in Eq. (5) for entropy minimization, the model may try to find a degenerate solution where the recovered sample $X_{Sl}^i = \mathcal{R}_\zeta(M(X_T^i))$ becomes constant across the masked region, as doing so can easily minimize entropy to zero.

Specifically, for a high masking ratio, $X_{Sl}^i$ is approximated by a constant value $c$, i.e.

$$X_{Sl}^i \approx c \quad \text{with } p_m \approx 1. \tag{6}$$

Passing this constant through the encoder results in constant feature representations:

$$\mathbf{z} = \mathcal{F}_\theta(X_{Sl}^i) \approx \mathcal{F}_\theta(c) = z_0. \tag{7}$$

In this case, the distribution of $\mathbf{z}$ collapses to a Dirac delta function centered at $z_0$:

$$p(\mathbf{z}) = \delta(\mathbf{z} - z_0). \tag{8}$$

By substituting Eq. (8) into the entropy (5) and using the property $\delta(\mathbf{x}) \log \delta(\mathbf{x}) = 0$ for a delta function $p(\mathbf{z}) = \delta(\mathbf{z} - z_0)$, we derive the entropy of the collapsed features:

$$H(p(\mathbf{z})) = - \int \delta(\mathbf{z} - z_0) \log \delta(\mathbf{z} - z_0) \, d\mathbf{z} = 0. \tag{9}$$

This implies that the entropy $H(p(\mathbf{z}))$ reaches its minimum value of zero, which satisfies the optimization objective but results in feature collapse. The model converges to a trivial solution where no meaningful variability in the recovered source-like sample exists.

**Conclusion** Given the high masking ratio, the recovery model $\mathcal{R}_\zeta$ is unable to generate a valid reconstruction of the source-like sample. Instead, it defaults to generating a constant value to minimize the entropy, resulting in collapsed features that carry no useful information. This trivial solution, characterized by $p(\mathbf{z}) = \delta(\mathbf{z} - z_0)$, leads to zero entropy, but the recovered sample fails to capture the temporal dependencies required for successful domain adaptation. In contrast, a lower masking ratio provides the recovery model with sufficient context, allowing for more meaningful reconstructions. When paired with our designed anchor-based recovery diversity maximization module, this results in diverse, temporally coherent recovered samples. Thus, a lower masking ratio, in conjunction with diversity-enhancing techniques, is critical to ensuring effective recovery and adaptation. (Sec 3.4 Enhancement.)

A.2 IMPROVED DIVERSITY WITH RECOVERY DIVERSITY MAXIMIZATION

**Theorem 2** *Maximizing the distance between original samples $\mathbf{X}_T^i$ and recovered samples $\mathbf{X}_{Sl}^i$ enhances the diversity of the recovered samples.*

**Proof:**

**Given Conditions**

- $X_T^i$ is a time-series sample from the target domain.
- $X_{Sl}^i$ is the corresponding recovered sample, generated by the recovery model $\mathcal{R}_\zeta$, i.e., $X_{Sl}^i = \mathcal{R}_\zeta(M(X_T^i))$, where $M(X_T^i)$ is the masked version of $X_T^i$.
- $p(X_T^i, X_{Sl}^i)$ denotes the joint probability distribution of the original samples $X_T^i$ and recovered samples $X_{Sl}^i$.
- $d(X_T^i, X_{Sl}^i)$ is the distance between the original and recovered samples.

**Conditional Entropy and Diversity**    The conditional entropy $H(X_{Sl}^i|X_T^i)$ measures the uncertainty in the recovered samples $X_{Sl}^i$, given the original samples $X_T^i$. As $X_{Sl}^i = \mathcal{R}_\zeta(M(X_T^i))$, higher conditional entropy implies greater uncertainty of $X_{Sl}^i$ generated from $X_T^i$, suggesting a wider range of possible outcomes for the recovered samples from their original samples. Therefore, increasing the conditional entropy directly corresponds to enhancing the diversity of the recovered samples.

**Conditional Entropy Equation**    The conditional entropy $H(X_{Sl}^i|X_T^i)$ quantifies the uncertainty in $X_{Sl}^i$, given $X_T^i$, and is defined as:

$$H(X_{Sl}^i|X_T^i) = -\sum_{X_T^i} \sum_{X_{Sl}^i} p(X_T^i, X_{Sl}^i) \log p(X_{Sl}^i|X_T^i). \tag{10}$$

This equation measures how much uncertainty remains in $X_{Sl}^i$ after observing $X_T^i$. Higher values of $H(X_{Sl}^i|X_T^i)$ indicate greater diversity in the recovered samples.

**Probability Decay with Distance**    We now show that the joint probability $p(X_T^i, X_{Sl}^i)$ is inversely related to the distance $d(X_T^i, X_{Sl}^i)$. Intuitively, nearby events have higher probabilities, while distant events have lower probabilities.

For example, in a Gaussian distribution, the probability density decays as the distance between $X_T^i$ and $X_{Sl}^i$ increases. Specifically:

$$p(X_T^i, X_{Sl}^i) \propto \exp\left(-\frac{d(X_T^i, X_{Sl}^i)^2}{2\sigma^2}\right). \tag{11}$$

Here, $d(X_T^i, X_{Sl}^i)$ is the distance between the original and recovered samples, and $\sigma^2$ is the variance. As $d(X_T^i, X_{Sl}^i)$ increases, the probability $p(X_T^i, X_{Sl}^i)$ decays exponentially.

Since the joint probability $p(X_T^i, X_{Sl}^i)$ decreases as $d(X_T^i, X_{Sl}^i)$ increases, the conditional entropy $H(X_{Sl}^i|X_T^i)$ from Eq. (10) also increases, indicating the uncertainty in $X_{Sl}^i$, given $X_T^i$, increases.

**Conclusion**    Maximizing the distance $d(X_T^i, X_{Sl}^i)$ decreases the joint probability $p(X_T^i, X_{Sl}^i)$, thus increasing the uncertainty and, therefore, the conditional entropy $H(X_{Sl}^i|X_T^i)$. As higher conditional entropy corresponds to greater diversity in the recovered samples, we conclude that maximizing the distance between the original and recovered samples enhances the diversity of the recovered distribution. (Sec 3.4 Enhancement.)

### A.3    Discussion of GANs for Source-like Domain Initialization

GAN-based works fail to meet the two outlined requirements for two reasons:

1. GANs typically use a random noise vector sampled from a standard distribution (e.g., Gaussian or uniform) as the initial input to the generator. This random initialization normally diverges significantly from the source distribution, expanding the solution space and making it challenging to converge to an optimal source-like distribution.

2. GANs are not inherently designed to handle sequential data or temporal dependencies, as they treat each generated sample independently without enforcing continuity between data points, so it may generate the random time points and fail to capture the temporal coherence.

## A.4 DATASET DETAILS AND PROCESSINGS

### A.4.1 UCI-HAR DATASET

The UCI-HAR dataset is tailored for human activity recognition tasks, comprising sensor data collected from 30 distinct users, each representing a separate domain. Each participant performs six activities: walking, walking upstairs, walking downstairs, standing, sitting, and lying down. The data is recorded using three types of sensors—accelerometers, gyroscopes, and body sensors—each capturing data on three axes. Thus, there are totally nine channels per sample, with each channel containing 128 data points. Following prior research (Ragab et al., 2023a), we employed a window size of 128 for sample extraction and applied min-max normalization for data preprocessing.

### A.4.2 SLEEP-EDF DATASET

The Sleep-EDF dataset is designed for sleep stage classification. It includes recordings from six channels monitoring various physiological signals, such as EEG (Epz-Cz, Pz-Oz), EOG, and EMG. Based on prior research (Ragab et al., 2023b) and due to the high information content in the Epz-Cz channel, we utilized only this channel in our experiments. The dataset comprises recordings from 20 subjects, each is treated as a domain because different persons have various personal habits. Each subject can be classified into five sleep stages: wake, light sleep stage 1 (N1), light sleep stage 2 (N2), deep sleep stage 3 (N3), and rapid eye movement (REM) (Goldberger et al., 2000). Notably, each sample in the dataset corresponds to a 30-second window of physiological data, recorded at a sampling rate of 100 Hz, resulting in 3000 timestamps per sample.

### A.4.3 MFD DATASET

The MFD dataset, collected by Paderborn University, is used for machine fault diagnosis, where vibration signals are leveraged to identify different types of incipient faults. Data was collected under four distinct working conditions, each treated as a separate domain. Each sample consists of a single univariate channel containing 5120 data points. (Sec 4.1 Datasets.)

## A.5 MODEL DETAILS

In our study, we adopted the encoder architecture presented in previous works (Ragab et al., 2023b;a), which is a 1-dimensional Convolutional Neural Network (CNN) comprising three layers with filter sizes of 64, 128, and 128, respectively. Each convolutional layer is followed by a Rectified Linear Unit (ReLU) activation function and batch normalization.

In the adaptation stage, we apply masking to generate masked samples, adopting a masking ratio of 1/8 across all datasets. To recover the masked samples, we designed a recovery model $\mathcal{R}_\zeta$, achieved by a two-layer Long Short-Term Memory network. The hidden dimension is set to 64 for the HAR and SSC tasks, and 128 for the MFD task, due to the longer time sequences in the latter. To generate anchor samples, we used an anchor ratio of 0.3 for all datasets, meaning the 30% of samples with the lowest entropy in the anchor bank are selected as anchor samples. For the temperature factor in Eq. (3) to achieve better anchor-based recovery diversity maximization, we used 0.05 for the MFD and EEG tasks, and 0.01 for the HAR task. (Sec 4.1 Unified Training Scheme.)

Table 5: Model parameters for baselines and ours.

| Models | HAR | | | | SSC | | | | MFD | | | |
|---|---|---|---|---|---|---|---|---|---|---|---|---|
| | Batch Size | Epochs | Pretrain LR | Adapt LR | Batch Size | Epochs | Pretrain LR | Adapt LR | Batch Size | Epochs | Pretrain LR | Adapt LR |
| DANN | 32 | 40 | - | 1e-2 | 32 | 40 | - | 5e-4 | 32 | 40 | - | 5e-4 |
| CDAN | 32 | 40 | - | 1e-2 | 32 | 40 | - | 1e-3 | 32 | 40 | - | 1e-3 |
| CoDATS | 32 | 40 | - | 1e-3 | 32 | 40 | - | 1e-2 | 32 | 40 | - | 5e-4 |
| CLUDA | 32 | 40 | - | 1e-3 | 32 | 40 | - | 1e-3 | 32 | 40 | - | 1e-3 |
| RAINCOAT | 32 | 50 | - | 5e-4 | 128 | 40 | - | 2e-3 | 32 | 40 | - | 1e-3 |
| SHOT | 32 | 40 | 1e-3 | 1e-4 | 32 | 40 | 3e-3 | 1e-5 | 32 | 40 | 1e-3 | 1e-5 |
| NRC | 32 | 40 | 3e-3 | 1e-5 | 32 | 40 | 3e-3 | 1e-5 | 32 | 40 | 1e-3 | 1e-5 |
| AaD | 32 | 40 | 3e-3 | 1e-4 | 32 | 40 | 3e-3 | 1e-5 | 32 | 40 | 1e-3 | 1e-5 |
| BAIT | 32 | 40 | 5e-4 | 1e-4 | 32 | 40 | 1e-3 | 5e-5 | 32 | 40 | 1e-3 | 1e-5 |
| MAPU | 32 | 40 | 1e-3 | 1e-4 | 32 | 40 | 3e-3 | 1e-5 | 32 | 40 | 1e-3 | 1e-5 |
| TemSR | 32 | 40 | 1e-3 | 5e-4 | 32 | 40 | 1e-3 | 5e-5 | 32 | 40 | 3e-3 | 7e-6 |

A.6 BASELINE DETAILS

We incorporate both conventional UDA approaches and source-free UDA (SFUDA) techniques, following prior work (Yang et al., 2022; Ragab et al., 2023b). Below is a summary of each baseline. Meanwhile, we provide the parameters used in each baseline, as shown in Table 5. (Sec 4.2 Comparisons with State-of-the-Arts)

**Conventional UDA methods**

- Domain-Adversarial Training of Neural Networks (DANN) (Ganin et al., 2016): DANN utilizes adversarial learning to push the encoder to generate domain-invariant features that a domain discriminator cannot tell which domain the sample comes from.

- Conditional Domain Adversarial Network (CDAN) (Long et al., 2018): CDAN leverages class-wise information with adversarial alignment for effective domain adaptation.

- Convolutional deep adaptation for time series (CoDATS) (Wilson et al., 2020): CoDATS uses adversarial learning to enhance adaptation performance, specifically targeting time-series data with limited supervision.

- Contrastive Learning-based Unsupervised Domain Adaptation (CLUDA): CLUDA leverages contrastive learning to capture contextual representations of time-series data, preserving label information and enabling domain-invariant alignment of contextual features across domains.

- fRequency-augmented AlIgN-then-Correct for dOmain Adaptation for Time series (RAIN-COAT): RAINCOAT tackles feature and label shifts by integrating both temporal and frequency features, aligning them across domains, and correcting misalignments to enhance the detection of domain-specific labels.

**Source-free UDA methods**

- Source Hypothesis Transfer (SHOT) (Liang et al., 2020): SHOT maximizes mutual information loss and employs self-supervised pseudo-labeling to extract target features aligned with the source hypothesis, enabling adaptation without requiring source data labels.

- Exploiting the intrinsic neighborhood structure (NRC) (Yang et al., 2021): NRC explores the underlying neighborhood structure in target data by forming distinct clusters and ensuring label consistency within them, addressing the challenge of unlabeled target domains.

- Attracting and dispersing (AaD) (Yang et al., 2022): AaD promotes consistent predictions within neighboring feature spaces, exploiting the intrinsic structure of unlabeled target data to improve adaptation.

- BAIT (Yang et al., 2023): BAIT uses a bait classifier to identify misclassified target features and subsequently updates the feature extractor to guide these difficult features toward the correct side of the decision boundary.

- Mask and impute (MAPU) (Ragab et al., 2023b): MAPU captures temporal dependencies in TS data by designing a temporal imputer in the source pretraining stage, and then restoring the temporal dependencies with the fixed imputer in the target adaptation stage for temporal dependency transfer.

A.7 INTUITIVE EXAMPLES FOR SEGMENT

Fig. 7 provides intuitive examples for generating segments from an recovered sample. Here, to intuitively illustrate masking parts and the extraction proportion of 4/6, the complete recovered sample is split into six portions. Fig. 7 (a) shows the complete version, with portions B, C, D, and E masked and recovered. Fig. 7 (b) demonstrates the extraction for the 'Early' segment, where portions A, B, C, and D are selected, capturing the information at the early stage of the sequence. Fig. 7 (c) shows the extraction for the 'Late' segment, selecting portions C, D, E, and F. Fig. 7 (d) shows the 'Recovered Parts' segment, where the portions containing recovered parts, including B, C, D, and E, have been extracted. (Sec 3.3 Optimization.)

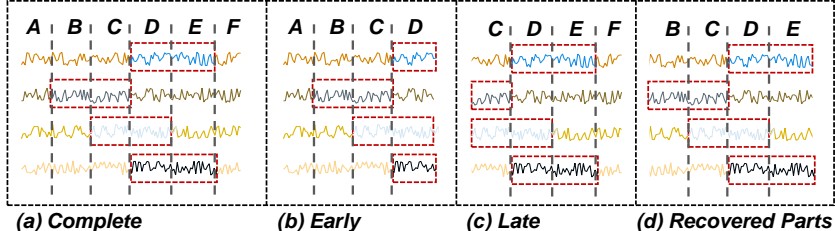

*(a) Complete*     *(b) Early*     *(c) Late*     *(d) Recovered Parts*

Figure 7: (a) The complete recovered sample. (b) (c) (d) Extracted segment for 'Early', 'Late', and 'Recovered Parts' containing four portions from different regions of the recovered sample.

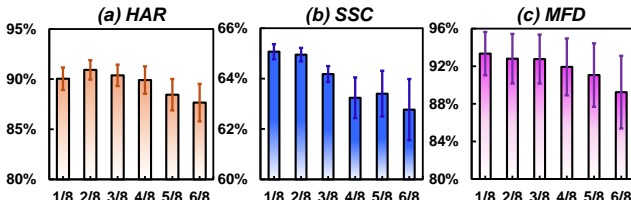

Figure 8: Analysis for Masking Ratio.

## A.8 ADDITIONAL RESULTS

Due to space limitations in the main paper, we here provide the analysis for the masking ratio, the anchor ratio, the extraction proportion, and the detailed results of the ablation study.

**Effect of Masking Ratio**   The masking ratio, which introduces diversity to the initial distribution for optimization as a source-like distribution, has been tested with values of [1/8, 2/8, 3/8, 4/8, 5/8, 6/8] following Ragab et al. (2023b), with larger values indicating more information removed in the sample. Fig. 8 shows the impact of various masking ratios, suggesting that smaller masking ratios lead to better performance. As discussed in Sec 3.4, while higher masking ratios introduce more diversity to the source-like distribution, they can cause the model to collapse by exploiting shortcuts, e.g., recovering the masked samples as a constant value. Although smaller masking ratios may limit diversity, our proposed recovery diversity maximization loss compensates for this by balancing the need for diversity with fidelity to the source domain. Thus, smaller masking ratios, e.g., 1/8 or 2/8, are recommended for achieving optimal results. (Sec 3.3 Masking and Recovery.)

**Effect of Anchor Ratio**   The anchor ratio, which determines the top-$k$ samples with the lowest entropy to generate the representative anchor, has been evaluated using [0.1, 0.3, 0.5, 0.7, 0.9], with larger values indicating more samples selected for generating the anchor sample. For example, 0.1 represents the 10% of samples with lowest entropy being selected for anchor generation. Fig. 9 shows the sensitivity of TemSR to different anchor ratios, where smaller anchor ratios tend to yield better results. This is because samples with the lowest entropy are more likely to produce high-quality anchors with greater confidence. In contrast, larger anchor ratios may include samples with lower confidence (with larger entropy), leading to less accurate anchors and, consequently,

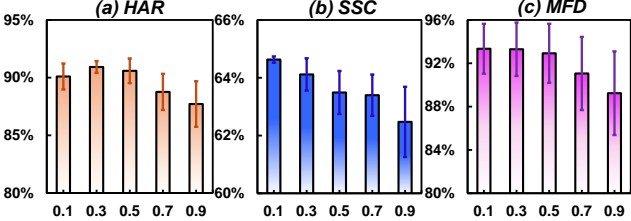

Figure 9: Analysis for Anchor Ratio.

poorer guidance during the adaptation process. From these results, anchor ratios of 0.1 or 0.3 are recommended for generating effective anchors to enhance performance. (Sec 3.4 Anchor Generation with Anchor Bank)

**Effect of Extraction Proportion** The extraction proportion determines the amount of local information in each segment. To evaluate its effectiveness, we tested the values within [7/8, 6/8, 5/8, 4/8, 3/8, 2/8]. A value of 1 represents segments containing only global information, while smaller values indicate that more local information is involved in entropy minimization. Fig. 10 presents the analysis of extraction proportions. From the figure, we observe that reducing the extraction proportion, e.g., from 7/8

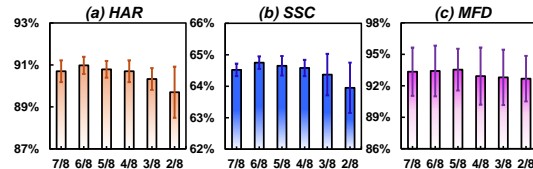

Figure 10: Analysis for extraction proportion.

to 6/8, can improve performance. This is because a lower proportion allows more local information to be included for entropy minimization, aligning the local distribution in recovered samples with the source distribution and thus achieving better source temporal recovery. However, with too small values, e.g., 2/8, each segment loses too much useful information from the recovered sample, making it hard to capture meaningful local dependencies. This leads the recovery model to misinterpret entropy minimization and produce ineffective source-like distributions, ultimately negatively impacting adaptation performance. Thus, an extraction proportion of 6/8 or 5/8 would be better for the optimization of the local distribution.

**Detailed Results for Ablation Study** The detailed results of the ablation study can be found in Tables 6, 7, and 8 for HAR, SSC, and MFD, respectively, further highlighting the importance of each module in generating a robust recovered source-like distribution for effective TS-SFUDA. (Sec 4.3 Ablation Study)

Table 6: Detailed ablation study of the ten HAR cross-domain scenarios regarding MF1 score (%).

| Variants | 2→11 | 12→16 | 9→18 | 6→23 | 7→13 | 18→27 | 20→5 | 24→8 | 28→27 | 30→20 | Avg. |
|---|---|---|---|---|---|---|---|---|---|---|---|
| Src-like Only | 26.39±9.04 | 27.33±9.20 | 09.76±5.91 | 12.75±4.38 | 18.87±5.64 | 19.45±10.6 | 21.61±7.41 | 12.57±5.56 | 19.45±0.32 | 8.63±1.48 | 17.68±7.89 |
| w/o $\mathcal{L}_{Seg}$ | 100.0±0.00 | 63.22±3.54 | 88.74±3.73 | 98.36±2.31 | 98.95±0.00 | 100.0±0.00 | 80.32±0.73 | 97.04±3.13 | 100.0±0.00 | 80.60±5.80 | 90.72±1.27 |
| w/o Early | 100.0±0.00 | 64.88±5.86 | 89.83±3.38 | 97.55±1.63 | 98.75±0.13 | 100.0±0.00 | 81.15±1.24 | 98.04±2.23 | 100.0±0.00 | 81.82±5.65 | 91.20±0.81 |
| w/o Late | 100.0±0.00 | 62.66±3.53 | 91.08±2.80 | 97.82±1.89 | 99.30±0.61 | 100.0±0.00 | 83.15±2.40 | 96.67±3.26 | 100.0±0.00 | 80.96±2.75 | 91.16±1.35 |
| w/o Recover | 100.0±0.00 | 66.87±5.42 | 90.31±4.11 | 96.73±0.00 | 98.31±0.06 | 100.0±0.00 | 82.73±1.43 | 100.0±0.00 | 100.0±0.00 | 84.35±5.47 | 91.93±0.95 |
| w/o Complete | 100.0±0.00 | 62.37±3.70 | 91.08±2.80 | 97.82±1.89 | 98.86±0.05 | 100.0±0.00 | 82.53±0.89 | 95.56±6.29 | 100.0±0.00 | 82.20±6.00 | 91.04±0.72 |
| w/o $\mathcal{L}_{SegSim}$ | 100.0±0.00 | 62.66±3.53 | 91.40±2.58 | 97.82±1.89 | 99.48±0.74 | 100.0±0.00 | 82.32±0.27 | 97.45±4.42 | 100.0±0.00 | 83.84±5.43 | 91.50±0.95 |
| w/o $\mathcal{L}_{ARDM}$ | 100.0±0.00 | 63.99±1.82 | 90.46±1.33 | 96.73±0.00 | 92.80±6.90 | 94.90±8.83 | 85.55±3.94 | 96.71±5.45 | 100.0±0.00 | 78.88±8.45 | 90.00±2.74 |
| w/o Anchor | 100.0±0.00 | 62.15±4.19 | 89.96±2.11 | 96.73±0.50 | 85.76±0.58 | 96.36±6.31 | 84.23±2.44 | 95.02±9.91 | 100.0±0.00 | 78.9±12.23 | 88.91±1.76 |
| w/o Add. Obj. | 99.45±0.78 | 61.33±3.03 | 83.95±1.08 | 98.37±2.31 | 96.14±3.97 | 100.0±0.00 | 82.55±0.75 | 96.97±6.06 | 100.0±0.00 | 82.55±8.84 | 90.13±1.68 |
| w/o Anchor Bank | 100.0±0.00 | 63.44±3.87 | 94.79±0.67 | 96.73±1.79 | 98.95±0.47 | 100.0±0.00 | 81.72±0.67 | 100.0±0.00 | 100.0±0.00 | 84.10±5.52 | 91.97±0.97 |
| w/ Random Init. | 100.0±0.00 | 62.80±4.74 | 93.26±4.92 | 97.82±1.89 | 98.95±0.07 | 100.0±0.00 | 82.40±1.11 | 100.0±0.00 | 100.0±0.00 | 84.52±4.53 | 91.97±0.92 |
| TemSR | 100.0±0.00 | 64.21±3.04 | 93.65±2.02 | 97.82±1.89 | 98.95±0.01 | 100.0±0.00 | 82.32±0.73 | 100.0±0.00 | 100.0±0.00 | 84.10±5.52 | 92.10±0.33 |

Table 7: Detailed ablation study of the ten SSC cross-domain scenarios regarding MF1 score (%).

| Variants | 16→1 | 9→14 | 12→5 | 7→18 | 0→11 | 3→19 | 18→12 | 13→17 | 5→15 | 6→2 | Avg. |
|---|---|---|---|---|---|---|---|---|---|---|---|
| Src-like Only | 13.54±5.76 | 13.74±3.17 | 11.45±1.63 | 11.49±0.49 | 33.65±2.71 | 22.77±18.0 | 09.87±2.21 | 03.85±2.56 | 06.22±0.00 | 07.83±4.41 | 13.44±2.07 |
| w/o $\mathcal{L}_{Seg}$ | 62.08±1.04 | 71.44±2.19 | 67.61±3.40 | 71.59±1.05 | 47.67±5.48 | 65.83±0.47 | 59.76±0.16 | 55.98±0.88 | 64.78±1.43 | 60.68±2.36 | 62.74±0.81 |
| w/o Early | 62.26±1.14 | 70.49±2.24 | 65.82±2.25 | 72.41±0.40 | 49.92±1.99 | 65.48±0.50 | 60.05±1.01 | 56.63±1.52 | 65.73±3.11 | 60.87±3.30 | 62.97±1.30 |
| w/o Late | 61.66±0.20 | 71.89±0.36 | 65.42±1.56 | 71.44±0.63 | 51.32±2.11 | 65.60±0.60 | 60.40±0.88 | 56.61±0.35 | 63.99±3.09 | 61.18±3.71 | 62.95±1.27 |
| w/o Recover | 62.26±1.14 | 72.44±0.75 | 66.40±2.13 | 71.54±0.40 | 49.84±2.07 | 65.68±0.59 | 60.72±0.98 | 56.68±2.05 | 66.40±1.51 | 62.66±1.73 | 63.46±0.37 |
| w/o Complete | 61.73±0.19 | 70.42±2.17 | 65.44±1.52 | 72.30±0.48 | 49.92±1.99 | 62.15±0.27 | 59.03±1.01 | 57.97±1.97 | 66.56±2.16 | 62.21±3.38 | 62.77±1.32 |
| w/o $\mathcal{L}_{SegSim}$ | 61.60±0.30 | 71.07±1.38 | 67.09±2.59 | 72.04±0.27 | 49.89±2.02 | 65.50±0.67 | 60.19±0.29 | 56.76±2.43 | 66.24±0.45 | 64.56±1.90 | 63.49±0.56 |
| w/o $\mathcal{L}_{ARDM}$ | 61.94±0.86 | 71.72±2.40 | 67.48±3.46 | 70.92±2.85 | 46.79±7.38 | 63.45±0.79 | 60.29±0.72 | 55.58±2.21 | 66.55±2.08 | 63.63±3.66 | 62.84±1.44 |
| w/o Anchor | 46.43±7.75 | 67.45±3.32 | 70.69±3.20 | 62.14±0.72 | 71.13±1.92 | 63.59±0.43 | 58.59±2.25 | 58.15±2.12 | 64.96±0.30 | 62.78±5.73 | 62.59±0.97 |
| w/o Add. Obj. | 62.72±1.05 | 71.70±2.51 | 65.14±3.66 | 71.76±1.52 | 47.92±5.72 | 66.72±0.62 | 60.05±0.62 | 57.21±2.50 | 66.46±0.33 | 64.64±2.10 | 63.43±0.50 |
| w/o Anchor Bank | 62.01±1.28 | 70.82±2.51 | 66.88±1.59 | 71.72±1.15 | 45.20±6.06 | 66.08±0.81 | 59.78±1.00 | 57.68±2.33 | 68.03±2.46 | 64.12±1.17 | 63.23±0.11 |
| w/ Random Init. | 62.04±1.24 | 70.68±1.98 | 66.84±1.72 | 72.00±1.18 | 47.56±5.14 | 65.49±0.09 | 61.66±2.29 | 58.23±1.76 | 64.64±0.46 | 64.49±1.37 | 63.36±0.20 |
| TemSR | 62.51±1.09 | 72.60±0.74 | 66.70±1.91 | 72.15±1.01 | 49.62±1.88 | 65.87±0.53 | 60.32±0.97 | 57.56±2.07 | 66.50±2.07 | 64.82±1.78 | 63.86±0.58 |

## A.9 COMPUTATION COMPLEXITY ANALYSIS

Model complexity analysis is crucial for assessing the practicality of TS-SFUDA techniques for real-world applications. As all methods utilize the same backbone, conducting a complexity anal-

Table 8: Detailed ablation study of the ten MFD cross-domain scenarios regarding MF1 score (%).

| Variants | 0→1 | 1→0 | 1→2 | 2→3 | 3→1 | 0→3 | 1→3 | 2→1 | 3→0 | 3→2 | Avg. |
|---|---|---|---|---|---|---|---|---|---|---|---|
| Src-like Only | 15.75±8.83 | 20.85±0.00 | 20.85±0.00 | 15.74±8.82 | 15.75±8.83 | 20.31±0.23 | 16.82±0.94 | 20.81±0.93 | 20.84±4.01 | 25.14±0.31 | 19.29±4.66 |
| w/o $\mathcal{L}_{Seg}$ | 99.96±0.06 | 85.82±4.89 | 82.57±5.34 | 94.65±3.34 | 99.98±0.03 | 91.88±0.06 | 100.0±0.00 | 96.02±0.11 | 88.52±0.74 | 83.88±0.50 | 92.33±2.15 |
| w/o Early | 99.76±0.24 | 86.68±3.73 | 82.92±4.50 | 95.18±4.11 | 100.0±0.00 | 96.92±2.11 | 100.0±0.00 | 94.52±3.21 | 85.52±0.38 | 80.24±2.65 | 92.17±2.47 |
| w/o Late | 99.86±0.24 | 86.56±3.70 | 81.36±6.86 | 97.59±2.90 | 100.0±0.00 | 97.78±1.12 | 100.0±0.00 | 95.09±5.44 | 87.78±0.43 | 83.51±10.1 | 92.95±3.19 |
| w/o Recover | 99.75±0.35 | 88.70±0.04 | 80.77±5.35 | 96.30±4.89 | 99.96±0.06 | 99.62±0.12 | 100.0±0.00 | 97.23±0.29 | 87.25±0.62 | 79.95±0.45 | 92.96±0.15 |
| w/o Complete | 99.86±0.24 | 86.59±3.73 | 80.40±5.19 | 97.48±4.09 | 100.0±0.00 | 96.79±0.00 | 100.0±0.00 | 94.88±5.15 | 85.44±1.05 | 81.28±0.53 | 92.27±3.35 |
| w/o $\mathcal{L}_{SegSim}$ | 99.70±0.26 | 86.60±3.74 | 84.49±5.95 | 95.26±4.04 | 99.92±0.14 | 99.92±0.01 | 100.0±0.00 | 96.24±3.33 | 87.18±0.74 | 80.09±8.43 | 92.94±2.67 |
| w/o $\mathcal{L}_{ARDM}$ | 85.31±5.31 | 85.88±6.04 | 85.99±3.54 | 95.20±3.87 | 99.97±0.05 | 99.92±0.08 | 100.0±0.00 | 96.75±3.93 | 88.51±0.53 | 87.06±8.16 | 92.46±2.41 |
| w/o Anchor | 85.38±5.33 | 82.13±3.77 | 99.97±0.05 | 83.52±5.29 | 95.20±3.87 | 99.85±0.02 | 100.0±0.00 | 95.68±2.57 | 87.54±0.54 | 85.67±6.28 | 91.49±2.34 |
| w/o Add. Obj. | 98.06±0.48 | 85.43±4.70 | 83.00±5.38 | 95.68±3.80 | 99.94±0.10 | 99.98±0.02 | 100.0±0.00 | 96.67±3.54 | 86.78±1.77 | 82.36±6.74 | 92.79±3.24 |
| w/o Anchor Bank | 100.0±0.00 | 86.92±3.95 | 80.15±4.76 | 97.53±2.92 | 100.0±0.00 | 99.95±0.05 | 100.0±0.00 | 96.33±3.54 | 87.65±1.44 | 82.33±6.47 | 93.09±2.32 |
| w/ Random Init. | 86.73±6.40 | 86.17±4.30 | 87.15±6.26 | 94.66±3.46 | 99.98±0.04 | 99.72±0.04 | 100.0±0.00 | 95.68±3.52 | 86.23±1.54 | 80.89±6.42 | 91.72±2.54 |
| TemSR | 99.97±0.05 | 87.03±4.05 | 84.47±5.88 | 95.23±3.85 | 100.0±0.00 | 99.95±0.05 | 100.0±0.00 | 96.67±4.21 | 87.17±1.56 | 81.96±5.09 | 93.24±1.83 |

ysis using standard metrics such as FLOPs or the number of parameters becomes challenging. Instead, considering that each method involves distinct operations during the adaptation stage, which can influence runtime, we compare their computational complexity by measuring the running time. Specifically, each method is executed once across all cross-domain cases on an RTX 3080Ti GPU. To ensure fairness, the analysis focuses exclusively on source-free UDA techniques.

From the results in Table 9, we observe that traditional methods generally require less time as they lack additional operations for recovering source temporal dependencies, which contributes to their poorer performance. While MAPU and TemSR incorporate additional operations, the extra runtime required is minimal (only a few seconds). Notably, compared to MAPU, TemSR does not rely on specific pretraining steps, thus resulting in reduced runtime overall. This demonstrates that TemSR not only effectively recovers temporal dependencies during the adaptation stage but also achieves this with limited computational resources, ensuring practical applicability.

Table 9: Running time comparisons of TS-SFUDA techniques.

| Models | SHOT | NRC | AaD | BAIT | MAPU | TemSR |
|---|---|---|---|---|---|---|
| Training Time/s | 70.43 | 66.63 | 72.94 | 74.36 | 89.79 | 83.24 |

## A.10 PSEUDOCODE OF OVERALL ADAPTATION PROCESS

The pseudo-code can be found in Algorithm 1, showing the training process of TemSR. (Sec 3.2 Overall Framework.)

**Algorithm 1** Overall Adaptation Process

```
# X_T, target sample [N, L], N: number of sensors, L: time length

# M: masking function
# H: entropy computation function

# F_S: source domain pretrained encoder
# G: source domain pretrained classifier

# F_T: target domain encoder, initialized by F_S
# R: recovery model

# A_B: anchor bank storing recovered samples
# E_B: entropy bank storing entropy values for recovered samples

# num_epochs: number of training epochs

F_S.eval() # Freeze source encoder
G.eval() # Freeze source classifier
F_T.train() # Trainable target encoder
R.train() # Trainable recovery model

# Initialize anchor and entropy banks
A_B.initial()
E_B.initial()

for epo in num_epochs:

    # Step 1: Masking and recovery
    X_hat = M(X_T) # Mask the target sample
    X_Sl = R(X_hat) # Recover masked target sample

    # Step 2: Update anchor and entropy banks
    E_Sl = H(G(F_S(X_Sl))) # Compute entropy of recovered sample
    A_B.update(X_Sl.detach()) # Update anchor bank with recovered samples
    E_B.update(E_Sl.detach()) # Update entropy bank

    # Step 3: Compute anchor-based recovery diversity maximization (L_ARDM)
    A = A_B.index(top_k(E_B)) # Select top samples by entropy
    L_ARDM = Anchor_Info_Max(X_Sl, X_T, A)

    # Step 4: Compute segment-based entropy loss (L_Seg)
    L_Seg = Segment_Entropy(X_Sl)

    # Step 5: Compute feature alignment loss (L_Align)
    h_Sl = F_S(X_Sl) # Extract features of source-like samples
    h_T = F_T(X_T) # Extract features of target samples
    L_Align = Alignment(h_Sl, h_T) # Align source-like and target features

    # Step 6: Compute target entropy loss (L_TrgEnt)
    L_TrgEnt = H(G(h_T)) # Compute entropy of target prediction

    # Step 7: Cycle between source-like optimization and adaptation
    if epo in source-like optimization phase:
        loss = combine_losses(L_ARDM, L_Seg, L_TrgEnt) # Source-like optimization
    else:
        loss = combine_losses(L_Align, L_TrgEnt) # Adaptation

    # Step 8: Backpropagation and optimization
    loss.backward()
    optimizer.step()
```

