# OpenReview forum: "Temporal Source Recovery for Time-Series Source-Free Unsupervised Domain Adaptation"
_ICLR.cc/2025/Conference — Submitted to ICLR 2025_

### Official Review · Reviewer_9pzb · 2024-10-24

**Soundness:** 2
**Presentation:** 3
**Contribution:** 3
**Rating:** 5
**Confidence:** 5

**Summary:**

The paper a noval framework, TemSR, for Time-Series Source-Free Unsupervised Domain Adaptation (TS-SFUDA), designed to recover and transfer temporal dependencies without access to source data. TemSR employs a masking, recovery, and optimization process to generate a source-like distribution with restored temporal dependencies. To improve the recovery process, segment-based regularization is introduced to capture local dependencies and maximize anchor-based recovery diversity to ensure diversity in the recovered distribution. Extensive experiments across various time-series tasks demonstrate the effectiveness of TemSR, surpassing existing methods that rely on source-specific pretraining designs while maintaining source data privacy.

**Strengths:**

1. The proposed task ensures source data privacy by performing domain adaptation without needing the source data.
2. The paper is logically structured, with a smooth connection between the motivation and the proposed method. Mathematical proofs are provided.
3. The introduction of the anchor-based recovery diversity maximization helps generate more diverse and realistic source-like distributions, improving adaptation performance.
4. 3 Benchmark datasets have been applied for evaluations.

**Weaknesses:**

1. The notation in Figure 1 is inconsistent with the notation in the text. Specifically, is the **L_sse** in Figure 1 the same as **L_seg** mentioned in line 259?
2. The non-source-free baseline models used are outdated, with CoDATS being published four years ago. It would be beneficial to include more recent baseline models from 2023, such as CLUDA ("Contrastive Learning in Unsupervised Domain Adaptation for Semantic Segmentation", Neurips 2023) and RainCoat ("Domain Adaptation for Time Series Under Feature and Label Shifts", ICML 2023).
3. Although using three benchmark datasets is sufficient, most previous MTS-UDA tasks typically select 7-10 source-target pairs (as the source-pair combination for HAR and SSC datasets can be hundreds) and report the average and standard deviation for comparisons, which provides a more balanced and fair evaluation. Moreover, the standard deviation across all pairs can demonstrate the model's robustness by highlighting its performance consistency across different cases. Having only five pairs per dataset is insufficient for a fair comparison.

**Questions:**

Methodologies:
1. In Section 3.3 on Recovery-initialization, could you further explain which parts are initialized in Figure.1 framework? Furthermore, could you provide further clarification on why initializing with the target distribution satisfies the first requirement:  "The initialized distribution should be close to the source distribution; otherwise, obtaining an effective source-like distribution is difficult". If possible, could you add an ablation study by initializing randomly? Could you further explain why GAN does not meet the two requirement?
2. Targeting to the "masking", as there is no "dimension related" information provided, could you give more details on the masking strategy? Will the masked parts be replaced with 0 or some noise? Or, you apply a similar masking strategy as MAE (Masked Autoencoders Are Scalable Vision Learners)?

Experiments:
1. In Section 4.1 (Datasets and Settings), you state that you followed the "identical training configurations as Ragab et al. (2023b)," referring to the paper titled "Source-Free Domain Adaptation with Temporal Imputation for Time Series Data. (MAPU)" However, after reviewing both the cited paper and the appendix of your work, I noticed that the datasets were split into training and testing sets only, with no mention of a validation set. If the best model is selected based on the test dataset, this could lead to biased evaluations, which raises concerns about the fairness of the evaluation process. In addition, For reproducibility, could you please include a table in the appendix listing key hyperparameters (e.g. learning rates, batch sizes, number of epochs) for all models, including baselines, to ensure fair comparison which could be easier for reading?
2. In the L_seg, four masking positions are utilized under the assumption that they can effectively capture local dependencies. Could you conduct some ablation study experiments to demonstrate that all four types of masking positions are significant, or how does each position affects the overall performance?
3. Could you add a computational complexity analysis with the proposed model and the baseline models?

---

> ### Author Response · Authors · 2024-11-21
> **Response [1/3]**
>
> **Weak 1 (W1).** The notation in Figure 1 is inconsistent with the notation in the text. Specifically, is the L_sse in Figure 1 the same as L_seg mentioned in line 259?
>
> **AWS:** Sorry for the mistakes. L_sse is L_Seg, and we have revised this in the manuscript.
>
> **W2.** The non-source-free baseline models used are outdated. W3. Insufficient source-target pairs.
>
> | Algorithm | 2$\rightarrow$11 | 12$\rightarrow$16 | 9$\rightarrow$18 | 6$\rightarrow$23 | 7$\rightarrow$13 | 18$\rightarrow$27 | 20$\rightarrow$5 | 24$\rightarrow$8 | 28$\rightarrow$27 | 30$\rightarrow$20 | AVG          |
> |-----------|------------------|-------------------|------------------|------------------|------------------|-------------------|------------------|------------------|-------------------|-------------------|--------------|
> | CLUDA     | 80.33%±3.81%     | 66.67%±2.24%      | 70.35%±2.13%     | 91.14%±1.70%     | 95.28%±2.62%     | 100.0%±0.00%      | 80.73%±3.24%     | 91.67%±3.15%     | 98.96%±1.47%      | 80.43%±2.34%      | 85.55%±1.24% |
> | RainCoat  | 100.0%±0.00%     | 76.28%±3.18%      | 77.35%±3.70%     | 98.14%±1.20%     | 100.0%±0.00%     | 100.0%±0.00%      | 85.73%±3.02%     | 97.67%±2.31%     | 100.0%±0.00%      | 86.46%±1.04%      | 92.16%±0.83% |
> | MAPU      | 100.0%±0.00%     | 67.96%±4.62%      | 82.77%±2.54%     | 97.82%±1.89%     | 99.29%±1.22%     | 100.0%±0.00%      | 82.88%±3.68%     | 96.48%±3.09%     | 96.01%±3.19%      | 85.43%±3.84%      | 90.86%±0.98% |
> | TemSR     | 100.0%±0.00%     | 64.21%±3.04%      | 93.65%±2.02%     | 97.82%±1.89%     | 98.95%±0.01%     | 100.0%±0.00%      | 82.32%±0.73%     | 100.0%±0.00%     | 100.0%±0.00%      | 84.10%±5.52%      | 92.10%±0.33% |
>
> **AWS:** 1. We have included comparisons with CLUDA and RainCoat for the HAR dataset. Due to time constraints, experiments on additional datasets are still in progress and will be reported in the camera-ready version. From the results, we observe that these recent methods, particularly RainCoat, demonstrate strong performance. However, it is important to note that **RainCoat is a traditional UDA method that requires access to source data during adaptation. In contrast, TemSR operates in a source-free setting and still achieves comparable performance, underscoring its effectiveness despite the stricter constraints.**
>
> 2. To address the concern regarding the number of cross-domain cases, we have conducted experiments on five additional cross-domain pairs for the HAR dataset, following the setting in [ref TKDD 2023]. Due to time limitations, these results are currently limited to MAPU as the baseline. Further experiments on other datasets and with more comprehensive baselines will be included in the camera-ready version. From these additional evaluations, TemSR continues to demonstrate its effectiveness, further validating the robustness and adaptability of our approach across diverse cross-domain scenarios.

---

> ### Author Response · Authors · 2024-11-21
> **Response [2/3]**
>
> **Methodologies Questions:**
>
>  **Q1.** Regarding Recovery-initialization, which parts are initialized in Figure.1 framework? Why initializing with the target distribution satisfies the first requirement. Ablation study by initializing randomly? Why GAN does not meet the two requirement?
>
> **AWS:** The masked data in Fig. 1 is initialized and masked from the target data. If the initial distribution significantly diverges from the source, aligning the two distributions becomes far more challenging. **Initializing with a highly random or distant distribution forces the recovery model to search through a much larger solution space** to approximate the source-like distribution. This expanded search space not only increases the complexity of finding an optimal solution but also heightens the risk of converging on a sub-optimal solution. A closer initial alignment with the source distribution narrows the search space, allowing the model to more efficiently and accurately capture essential source characteristics, leading to a more effective source-like domain.
>
> We have also included **an ablation study with random initialization** to evaluate its impact. The results show that this random initialization for the source-like domain significantly reduces performance. Furthermore, the standard deviation increases, highlighting the greater difficulty in identifying an optimal solution.
>
> | Variants                              | HAR          | SSC          | MFD          |
> |---------------------------------------|--------------|--------------|--------------|
> | w/ random initialization for src-like | 90.57%±2.25% | 63.82%±1.49% | 90.94%±7.68% |
> | TemSR                                 | 90.93%±0.54% | 64.72%±0.19% | 93.34%±2.31% |
>
> Regarding GANs, it fail to meet the two outlined requirements for two reasons: 1. GANs typically use a random noise vector sampled from a standard distribution (e.g., Gaussian or uniform) as the initial input to the generator. This random initialization normally diverges significantly from the source distribution, expanding the solution space and making it challenging to converge to an optimal source-like distribution. 2. GANs are not inherently designed to handle sequential data or temporal dependencies, as they treat each generated sample independently without enforcing continuity between data points, so it may generate the random time points and fail to capture the temporal coherence.
>
>
>
>  **Q2.** More details on the masking strategy? Similar with MAE?
>
> **AWS:** To clarify the masking strategy, suppose we use a masking ratio of 3/8. In this case, we divide each TS data into 8 segments, and 3 of these segments are randomly selected for masking by setting their values to 0. For multivariate TS data, where multiple channels are present, we apply this masking process independently to each channel.
>
> Regarding MAE, this technique aims to mask and reconstruct the original samples, while our approach focuses on masking and recovering the masked samples toward a source-like distribution. **As the source-like samples are not available for direct reconstruction, we cannot use reconstruction loss as in MAE. Instead, we introduce a regularization loss L_Seg** to guide the recovery process, encouraging the masked samples to approximate the source-like distribution.

---

> > ### Author Response · Authors · 2024-11-21
> > **Response [3/3]**
> >
> > **Experiments Questions:**
> >
> >  **Q1.** No mention of a validation set. A table listing key hyperparameters?
> >
> > **AWS:** As the baseline results are directly adopted from MAPU, we followed the identical training configurations outlined in their work to **ensure fair comparisons**. Consistent with MAPU, the best model is selected based on **the final epoch of training on the training dataset, without the use of a separate validation set**. This approach avoids introducing additional datasets while maintaining alignment with the referenced study for fair comparisons. For reproducibility, we have included a table in the appendix detailing the key hyperparameters for all models in the camera ready version.
> >
> >  **Q2.** Ablation study to demonstrate the effects of four types of masking positions.
> >
> > | Variants           | HAR          | SSC          | MFD          |
> > |--------------------|--------------|--------------|--------------|
> > | w/o Early          | 90.24%±1.00% | 64.18%±0.58% | 92.91%±2.28% |
> > | w/o Late           | 90.10%±1.06% | 64.34%±0.13% | 93.07%±1.15% |
> > | w/o Recover        | 90.57%±0.91% | 64.50%±0.04% | 93.10%±0.14% |
> > | w/o Complete       | 90.04%±1.13% | 63.96%±0.89% | 92.87%±2.27% |
> > | TemSR              | 90.93%±0.54% | 64.72%±0.19% | 93.34%±2.31% |
> >
> > **AWS:** We have expanded the ablation study to include the components of the segment entropy loss. The average results are summarized in the main text, while detailed cross-domain case results are included in the appendix. From the ablation study on the segment entropy loss, we observe that each component plays a critical role, as removing any of them results in performance degradation. Notably, the "complete" is the most significant, as it captures global dependencies. When this component is removed, only local dependencies are captured, which adversely affects the model's performance. Among the components targeting local dependencies, the early and late segments are relatively more impactful than the recovered segment. This is likely because the early and late segments sometimes intersect with the recovered parts. However, this does not diminish the importance of the recovered segment, as it focuses on the masked parts, encouraging them to align with source temporal dependencies and further enhancing performance.
> >
> >  **Q3.** Computational complexity analysis.
> >
> > **AWS:** As all methods utilize the same backbone, conducting a complexity analysis using standard metrics such as FLOPs or the number of parameters becomes challenging. Instead, considering that each method involves distinct operations during the adaptation stage, which can influence runtime, we compare their computational complexity by measuring the running time. Specifically, each method is executed once across all cross-domain cases on an RTX 3080Ti GPU. To ensure fairness, the analysis focuses exclusively on source-free UDA techniques.
> >
> > From the results in the table below, we observe that traditional methods generally require less time as they lack additional operations for recovering source temporal dependencies, which contributes to their poorer performance. While MAPU and TemSR incorporate additional operations, the extra runtime required is minimal (only a few seconds). Notably, compared to MAPU, TemSR does not rely on specific pretraining steps, thus resulting in reduced runtime overall. This demonstrates that TemSR not only effectively recovers temporal dependencies during the adaptation stage but also achieves this with limited computational resources, ensuring practical applicability.
> >
> > | Models          | BAIT  | SHOT  | AaD   | NRC   | MAPU  | TemSR |
> > |-----------------|-------|-------|-------|-------|-------|-------|
> > | Training Time/s | 74.36 | 70.43 | 72.94 | 66.63 | 89.79 | 83.24 |
> >
> > [TKDD 2023] AdaTime: A Benchmarking Suite for Domain Adaptation on Time Series Data

---

> > > ### Author Response · Authors · 2024-11-25
> > > **Response Follow-up**
> > >
> > > Dear Reviewer 9pzb,
> > >
> > > Thank you once again for taking the time to review our paper and provide valuable suggestions and comments. We would like to confirm whether the points addressed in our rebuttal are clear and aligned with your feedback. Please do not hesitate to let us know if any additional clarifications or details are needed.
> > >
> > > We greatly appreciate your time and consideration.

---

> > > > ### Comment · Reviewer_9pzb · 2024-11-26
> > > >
> > > > Most of my concerns have been resolved. However, further refinement is necessary for the manuscript to meet the acceptance standards. Additional source-target experiments should be included no later than this stage, as they are crucial for evaluating the method's fairness. Therefore, I will still maintain my score.

---

> > > > > ### Author Response · Authors · 2024-11-28
> > > > >
> > > > > Thank you for your valuable suggestions. We have conducted all the additional experiments and incorporated them into the updated version of the paper. These include:
> > > > >
> > > > > - Additional cross-domain scenarios (Tables 1, 2, and 3).
> > > > > - Comparisons with CLUDA and RainCoat (Tables 1, 2, and 3).
> > > > > - An extended ablation study (Table 4).
> > > > > - A comprehensive table listing all parameters (Table 5 in the Appendix).
> > > > > - Complexity analysis (Table 9 and Appendix A.9).
> > > > > - Detailed explanations on why GANs are unsuitable for our requirements (Appendix A.3).
> > > > >
> > > > > We appreciate your feedback and believe these updates address the points raised. Please let us know if you have any further concerns or suggestions.

---

> > > > > > ### Author Response · Authors · 2024-12-02
> > > > > >
> > > > > > Dear Reviewer 9pzb,
> > > > > >
> > > > > > We hope our response has clarified your concerns.
> > > > > >
> > > > > > Please let us know if anything remains unclear or requires further explanation. We’d greatly appreciate your thoughts.

---

> > > > > > > ### Author Response · Authors · 2024-12-03
> > > > > > >
> > > > > > > Dear Reviewer 9pzb,
> > > > > > >
> > > > > > > Thank you once again for your valuable suggestions regarding additional experiments. We have addressed your feedback in the **updated PDF**, with all revised parts highlighted in **blue**, including revised explanations and the newly added experiments.
> > > > > > >
> > > > > > > If you have any further concerns or suggestions, please let us know. Additionally, if you find that the revised experiments and explanations meet your expectations, we would greatly appreciate it if you could consider adjusting your score accordingly.
> > > > > > >
> > > > > > > Thank you for your time and thoughtful feedback.

---

### Official Review · Reviewer_5jCn · 2024-10-30

**Soundness:** 2
**Presentation:** 3
**Contribution:** 2
**Rating:** 3
**Confidence:** 4

**Summary:**

In this paper, the authors present a temporal source recovery (TemSR) framework for source-free unsupervised domain adaptation. Specifically, a source-like distribution is generated with the source temporal dependencies recovered. A segment-based regularization and an anchor-based recovery diversity maximization are developed to enhance the source recovery. Experiments on three datasets are done to verify the proposed TemSR.

**Strengths:**

1. The paper proposes to deal with a practical and meaningful adaptation scenario, that is source-free unsupervised domain adaptation.
2. The paper presents a temporal source recovery (TemSR) framework for source-free unsupervised domain adaptation.
3. The paper conduct experiments on three practical datasets, and demonstrate the effectiveness of the proposed TemSR  compared with given baselines.

**Weaknesses:**

1. One of my major concerns of the work is on the technical novelty. Using mask and then recovery has been shown the adaptation and transfer capability in generation tasks, e.g., masked autoencoder and videoMAE. The application of such an idea to source-free UDA setting makes the technical significance limited although it may not be explicitly done before.
2. The paper misses some important video-based UDA related works. Video is a popular time-series data, and there are quite a lot of video-based UDA research. The reviewer suggests the authors to discuss them in related work section.
3. For time-series data as stated in the setting with N channels and L time points, existing UDA works usually handle both the spatial and temporal divergence, e.g., [ref1,2]. However, the paper seems to overlook the spatial divergence and only focus on the temporal one.
4. The experimental studies are not very convincing. The author may refer to my detailed comments on the below questions.

**Questions:**

1. Figure 2 is not very informative. It is inadequate to use two ellipses to represent complex temporal and spatial divergence across TS data. Moreover, what does “target distribution” mean here? Are you trying to explain why using masked target data?
2. Regarding the recovery component, is a reconstruction loss included? Or using a pretrained reconstruction model as backbone? If not, is it better to initialize the recovery model by a reconstruction model pretrained on the target unsupervised data?
3. Regarding the segment regularization line 248-250, low entropy difference between segments does not indicate the smooth dependencies among segments. The authors may need to present ablation studies on this term (Eq. (2)) to show its effectiveness. Moreover, what does {C,E,L,R} mean? As in Eq. (2), (k, s) belongs to {C,E,L,R}, is it trying to minimize the entropy difference between arbitrary segment pairs?
4. Can you elaborate more on the test flow? Are you using the pretrained source classifier for final prediction?
5. Regarding the experiments parts: (1) more complex video benchmark datasets are encouraged to be tested; (2) why using the same CNN backbone? For different tasks, different backbones should be used. For instance, for action recognition tasks, there are more advanced backbones, e.g., I3D, videomae; (3) Model details should be presented in the main contents. Please elaborate more on the recovery model as it is the main contribution. Are you using LSTM or Bi-LSTM? Why not considering Transformer structure? (4) The results are not very convincing to show the performance superiority of TemSR as TemSR is the winner in 2 of 5, 1 of 5, and 3 of 5 tasks in the three datasets. (5) Sensitivity analysis shows that the optimal hyper-parameter value varies across tasks. It is not convincing to conclude the best value range is 1 to 10 by only using 3 datasets. The hyper-parameter generalization ability of the proposed TemSR is also an issue when facing new tasks or datasets. Moreover, what is “EEG” in figures 4,5,6?

[ref1] Contrast and mix: Temporal contrastive video domain adaptation with background mixing

[ref2] Unsupervised video domain adaptation for action recognition: A disentanglement perspective

---

> ### Author Response · Authors · 2024-11-21
> **Response [1/3]**
>
> **Weak 1 (W1).** Regarding technical novelty, mask and then recovery has been shown the adaptation and transfer capability in generation tasks, e.g., masked autoencoder and videoMAE. The application of such an idea to source-free UDA setting makes the technical significance limited although it may not be explicitly done before.
>
> **AWS:** We would like to highlight the novelty of our work in three key aspects, distinguishing it from MAE and its variants:
>
> 1. **Different focus**: MAE emphasizes masking and reconstructing the original sample, using the masking process as a core mechanism. In contrast, our work uses masking as a **tool**, with the primary focus on recovering masked samples as source-like samples **rather than reconstructing the original sample**. This shift in objective fundamentally changes the methodology and application.
>
> 2. **Different loss functions**: Due to differing aims, MAE relies on reconstruction losses, such as MSE, to align the reconstructed and original samples. However, our approach seeks to recover source-like samples, which do not explicitly exist. Therefore, **we use entropy minimization as a guiding principle**. By leveraging the property that models with minimized entropy produce deterministic outputs, we employ this constraint inversely to guide the recovery.
>
> 3. **Specialized design for time-series data**: Our work is specifically tailored for TS-SFUDA tasks, leveraging masking as a tool to capture and incorporate the intrinsic temporal dependencies of TS data for enhanced TS-SFUDA performance. To the best of our knowledge, this is **the first approach** to employ such a concept in the TS-SFUDA setting.
>
> **W2.** Missing video-based UDA related works, and video is a popular time-series data. **W3.** Overlook the spatial divergence and only focus on the temporal one.
>
> **AWS:** Video, time-series, and language data share similarities as sequential data, but they also exhibit distinct characteristics. For instance, video data is an extension of image data, with frames containing rich spatial and temporal information. Consequently, video-based UDA often emphasizes both spatial and temporal modeling. In contrast, TS data consists of **discrete limited data points at each time step**, so most TS-related works focus primarily on temporal dependencies [ref ICLR 2017, 2023, TPAMI 2023]. Furthermore, most TS-focused research [ref ICLR 2017, 2023, TPAMI 2023, 2024] does not explore the relevance of video data, as the emphasis typically lies on the unique properties of TS data. We will incorporate a discussion of relevant video-based UDA works in the related work section to provide a more comprehensive perspective.

---

> > ### Comment · Reviewer_5jCn · 2024-11-27
> > **Further comments.**
> >
> > I would appreciate the authors' detailed response to my reviews.
> > For W2 and W3, I am not very convinced as TS data is N*d-dimensional and the spatial divergence should exist, note that video data (may be some pre-processed visual features) is a special case. The reviewer understands that some existing works only consider the temporal divergence, but there also quite a lot works considering both.

---

> ### Author Response · Authors · 2024-11-21
> **Response [2/3]**
>
> **Q1.** Figure 2 inadequate to represent complex temporal and spatial divergence across TS data. What does “target distribution” mean here?
>
> **AWS:** The "target distribution" refers to the data distribution in the target domain, which we use as the initialization for generating the source-like domain. We acknowledge that due to the complex dependencies in time-series data, accurately depicting the divergence between domains with a simple diagram is challenging. However, **the figure is intended to convey a high-level conceptual understanding, aligning with the simplified visualizations commonly used in prior works (e.g., [ref ICML 2023, TPAMI 2024, CVPR 2023])** to illustrate key ideas. Specifically, our diagram illustrates that while the target domain differs from the source domain, the two domains are still related, and the target domain is not drastically divergent from the source domain. This relationship motivates to use the target distribution for initialization.
>
> **Q2.** Regarding the recovery component, is a reconstruction loss included? Or using a pretrained reconstruction model as backbone?
>
> **AWS:** In the recovery process, we design **a regularization loss L_Seg, rather than a reconstruction loss**, to guide the masked samples toward becoming source-like samples. As **our goal is not to reconstruct the original samples**, we argue that using a recovery model pretrained with target data may not provide significant benefits. To evaluate this, we conducted an experiment comparing the performance of a variant using a pretrained recovery model with our vanilla model. The results (**90.76 w/ pretrained vs. 90.93 w/o pretrained**, regarding the average performance) indicate that the pretrained recovery model does not provide a notable advantage in this context.
>
>
> **Q3.** Ablation studies on low entropy difference between segments to show its effectiveness. What does {C,E,L,R} mean?
>
> **AWS:** In Eq. (2), {C, E, L, R} represent the complete recovered sample (C), the early segment (E), the late segment (L), and the segment containing all recovered parts (R).
>
> The equation minimizes the entropy differences between these segment pairs to encourage consistent temporal dependencies. **The rationale for the smooth dependencies among segments is as follows:**
>
> 1. Entropy minimization for aligning with the source distribution:
>
> Models with minimized entropy can produce deterministic outputs, and this ideal output constraint can be inversely employed to guide adaptation. Thus, with the minimized entropy on source data, the source model can produce deterministic outputs for distributions with source characteristics. By minimizing the entropy computed by the fixed source model for recovered samples, this constraint can inversely regularize the samples, forcing them to align with the source distribution. Here, the recovery model is forced to capture source temporal dependencies, as only by understanding these dependencies can the model effectively reconstruct masked parts, minimize entropy, and ensure recovered samples align with the source distribution.
>
> 2. Minimizing Entropy Differences to Ensure Smooth Dependencies:
>
> **By minimizing entropy across segments, as per Eq. (1), and reducing the entropy differences between segment pairs (Eq. (2))**, the model ensures smooth temporal dependencies among segments. This consistency is crucial for recovering the global and local structure of TS data.
>
> We also conducted **ablation studies on L_SegSim**, showing that removing this loss results in performance drops across all three datasets. While not substantial, the results can also highlight the effectiveness of this loss in enhancing recovery and alignment.
>
>
> | Variants           | HAR          | SSC          | MFD          |
> |--------------------|--------------|--------------|--------------|
> | w/o L_SegSim       | 90.17%±1.00% | 64.34%±0.32% | 93.19%±2.39% |
> | TemSR              | 90.93%±0.54% | 64.72%±0.19% | 93.34%±2.31% |
>
> **Q4.** The test flow?
>
> **AWS:** After the adaptation stage, the optimization target encoder extracts features from the testing sample, which are then classified by the pretrained source classifier for final predictions.
>
> **Q5. (1)** Video benchmark datasets
>
> **AWS:** As discussed earlier, video data differs significantly from TS data, and TemSR is specifically designed for TS tasks, focusing on temporal dependencies unique to TS data, so it is hard to directly adapt TemSR to video tasks. Extending TemSR to video tasks will be considered in future work.

---

> ### Author Response · Authors · 2024-11-21
> **Response [3/3]**
>
> **Q5. (2)** why using the same CNN backbone?
>
> **AWS:** While task-specific backbones can improve performance, our goal is to evaluate TemSR’s effectiveness rather than the impact of the backbone. Therefore, we use a consistent backbone, as done in similar TS studies [ref ICML 2023, ICLR 2023, TPAMI 2024], to ensure **fair comparisons** and **attribute performance differences to the algorithm rather than the backbone**.
>
> **Q5. (3)** Model details, particularly on the recovery model. Why not considering Transformer structure?
>
> **AWS:** Due to space limitations, model details are included in the appendix, as is common in other works [ref ICML 2023, ICLR 2023, TPAMI 2024]. We use a two-layer LSTM for the recovery model due to its strong imputation ability for TS data [ref KDD 2023]. **We also evaluated a transformer-based recovery model. However, its performance was slightly lower, with average results of 89.21%±1.45%, 64.38%±0.39%, and 92.45%±3.07% on the HAR, SSC, and MFD datasets, respectively, compared to the two-layer LSTM**. This underperformance is attributed to the transformer's reliance on attention mechanisms to learn temporal dependencies. While effective, these mechanisms typically require larger datasets to fully capture patterns. Unlike video or image tasks, where abundant data points is often available, time-series data is typically limited, making transformers less effective in this context. More introduction regarding the recovery model will be provided in the camera ready version.
>
> **Q5. (4)** Not convincing results, as TemSR is the winner in 2 of 5, 1 of 5, and 3 of 5 tasks in the three datasets.
>
> **AWS:** The main advantage of TemSR is its ability to transfer temporal dependencies without requiring domain-specific designs during source pretraining, and it achieves competitive performance in such restricted cases. For example, in the SSC task, where TemSR wins only 1 of 5 tasks, MAPU, which relies on source-specific designs, achieves the best performance in most cases. However, MAPU’s dependency on such pretraining makes it less practical in real-world scenarios, while **TemSR achieves comparable performance without requiring any source specific designs**.
>
> **Q5. (5)** Not convincing sensitivity analysis by only using 3 datasets. The hyper-parameter generalization ability. What is “EEG” in figures 4,5,6?
>
> **AWS:** We apologize for the lack of clarity. The sensitivity analysis presented in Figures 4 and 5 reflects the average results across all cross-domain cases in each task, **encompassing a total of 15 cross-domain scenarios**. This broader evaluation provides a more comprehensive perspective, rather than being limited to only three datasets. Furthermore, **additional experiments conducted on extended cross-domain scenarios** on HAR reaffirm TemSR's consistent and strong performance compared to the second-best method, MAPU. Regarding "EEG" in Figures 4, 5, and 6, we apologize for the oversight. "EEG" refers to the SSC task, as the SSC dataset is based on EEG data collected for experiments. We have corrected this mislabeling in the revised version.
>
> | Algorithm | 2$\rightarrow$11 | 12$\rightarrow$16 | 9$\rightarrow$18 | 6$\rightarrow$23 | 7$\rightarrow$13 | 18$\rightarrow$27 | 20$\rightarrow$5 | 24$\rightarrow$8 | 28$\rightarrow$27 | 30$\rightarrow$20 | AVG          |
> |-----------|------------------|-------------------|------------------|------------------|------------------|-------------------|------------------|------------------|-------------------|-------------------|--------------|
> | MAPU      | 100.0%±0.00%     | 67.96%±4.62%      | 82.77%±2.54%     | 97.82%±1.89%     | 99.29%±1.22%     | 100.0%±0.00%      | 82.88%±3.68%     | 96.48%±3.09%     | 96.01%±3.19%      | 85.43%±3.84%      | 90.86%±0.98% |
> | TemSR     | 100.0%±0.00%     | 64.21%±3.04%      | 93.65%±2.02%     | 97.82%±1.89%     | 98.95%±0.01%     | 100.0%±0.00%      | 82.32%±0.73%     | 100.0%±0.00%     | 100.0%±0.00%      | 84.10%±5.52%      | 92.10%±0.33% |
>
>
> [**ICLR 2017**] VARIATIONAL RECURRENT ADVERSARIAL DEEP DOMAIN ADAPTATION
>
> [**ICLR 2023**] CONTRASTIVE LEARNING FOR UNSUPERVISED DOMAIN ADAPTATION OF TIME SERIES
>
> [**TPAMI 2023**] CALDA: Improving Multi-Source Time Series Domain Adaptation With Contrastive Adversarial Learning
>
> [**TPAMI 2024**] SEA++: Multi-Graph-Based Higher-Order Sensor Alignment for Multivariate Time-Series Unsupervised Domain Adaptation
>
> [**ICML 2023**] Domain Adaptation for Time Series Under Feature and Label Shifts
>
> [**CVPR 2023**] Guiding pseudo-labels with uncertainty estimation for source-free unsupervised domain adaptation
>
> [**KDD 2023**] Source-Free Domain Adaptation with Temporal Imputation for Time Series Data

---

> > ### Author Response · Authors · 2024-11-25
> > **Response Follow-up**
> >
> > Dear Reviewer 5jCn,
> >
> > Thank you once again for taking the time to review our paper and provide valuable suggestions and comments. We would like to confirm whether the points addressed in our rebuttal are clear and aligned with your feedback. Please do not hesitate to let us know if any additional clarifications or details are needed.
> >
> > We greatly appreciate your time and consideration.

---

> ### Author Response · Authors · 2024-11-28
>
> Thank you for your thoughtful feedback. We appreciate the opportunity to address your concerns regarding W2 and W3. While we acknowledge that some works consider both temporal and spatial divergences in their approaches, we would like to highlight two key points relevant to this work:
>
>
>
> 1. We focus on demonstrating the effectiveness of our idea
>
> The central goal of this work is to validate the effectiveness of our approach: **generating a source-like domain by recovering inherent dependencies within time-series (TS) data for TS-SFUDA**. Temporal dependencies are inherent in TS data, making them a universal feature across tasks. While introducing additional spatial information can enhance performance in specific tasks where extensive sensors are used, our focus here is on temporal dependencies to **establish a solid foundation** for our method. In future work, we plan to incorporate spatial information into TS data to further improve performance.
>
> 2. Generality through temporal dependency modeling
>
> Unlike video data, many TS tasks lack strong spatial characteristics, **making temporal dependency modeling a more universally applicable starting point**. For example, in scenarios where only one sensor or a single channel is used (e.g., due to cost constraints), no spatial information exists. Even when TS data is N×d-dimensional, the channels may represent diverse signals without meaningful spatial relationships. For instance:
>
> - In certain human activity recognition datasets [1], three channels are collected, representing the x, y, and z axes from a single sensor, which do not inherently encode spatial structure.
>
> - Similarly, in the UCI-HAR dataset used in this work, the nine channels correspond to signals from three different types of sensors (e.g., body acceleration and angular velocity), each comprising x, y, and z axes. These signals lack the clear spatial semantics observed in images or videos.
>
> By focusing on temporal dependencies, our method addresses a broader range of TS tasks and lays a strong foundation for future advancements.
>
> We hope this clarification addresses your concerns. Please let us know if you have any further feedback.
>
> [1] Activity recognition using cell phone accelerometers

---

> > ### Author Response · Authors · 2024-12-02
> >
> > Dear Reviewer 5jCn,
> >
> > We hope our response has clarified your concerns.
> >
> > Please let us know if anything remains unclear or requires further explanation. We’d greatly appreciate your thoughts.

---

> > > ### Author Response · Authors · 2024-12-03
> > > **Response Follow-up**
> > >
> > > Dear Reviewer 5jCn,
> > >
> > > Thank you once again for your valuable suggestions and follow-up questions regarding our work. We would like to kindly ask if our responses have adequately addressed your concerns. If so, we would greatly appreciate it if you could consider adjusting your score, as this would be highly significant for us.
> > >
> > > Thank you again for your time and thoughtful feedback.

---

### Official Review · Reviewer_gLQz · 2024-11-03

**Soundness:** 2
**Presentation:** 3
**Contribution:** 2
**Rating:** 6
**Confidence:** 3

**Summary:**

This paper addresses the time-series source-free domain adaptation problem. It proposes a source recovery method that recovers target data to a source-like distribution and introduces anchor-based recovery diversity maximization to enhance the diversity of the source-like distribution.

**Strengths:**

1. The overall idea of source distribution recovery is reasonable.
2. The method section is detailed and clearly presented.

**Weaknesses:**

1. The ablation study could be more comprehensive, as there are additional specific designs with intuitive explanations mentioned in the method section beyond the three variants in Table 4, such as the consistent segment entropy loss and different components in the ARDM loss.
2. The two key requirements stated at the beginning of Section 3.3 are intuitive. More explanation or support from the literature would be beneficial.

**Questions:**

In Section 4.5, could the authors explain why minimizing the domain discrepancy between the source-like and target domains is required for domain adaptation? This point does not appear to be discussed in previous sections.

---

> ### Author Response · Authors · 2024-11-21
> **Response [1/2]**
>
> **Weak 1 (W1).** More ablation study, such as the consistent segment entropy loss and the ARDM loss.
>
>
> **AWS**: We have expanded the ablation study to analyze the components of the segment entropy loss and the contributions of various elements in the ARDM loss. Specifically:
>
> - For the segment entropy loss, w/o Early - w/o Complete refer to the removal of the corresponding segments in Eq. (1), while w/o L_SegSim indicates the exclusion of the segment similarity loss.
>
> - For the ARDM loss, w/o Anchor denotes the removal of anchors, which also eliminates the additional objective since it is designed to enhance anchor guidance. Meanwhile, w/o Additional Obj refers to the exclusion of only the additional objective while retaining the anchors.
>
> From the ablation study on the segment entropy loss, we observe that each component plays a critical role, as removing any of them results in performance degradation. Notably, the "complete" is the most significant, as it captures global dependencies. When this component is removed, only local dependencies are captured, which adversely affects the model's performance. Among the components targeting local dependencies, the early and late segments are relatively more impactful than the recovered segment. This is likely because the early and late segments sometimes intersect with the recovered parts. However, this does not diminish the importance of the recovered segment, as it focuses on the masked parts, encouraging them to align with source temporal dependencies and further enhancing performance. Additionally, removing L_SegSim results in a noticeable drop in performance, highlighting the effectiveness of maintaining the consistent temporal dependencies across these segments.
>
> Regarding the ARDM loss, removing the anchor results in a significant performance decline due to insufficient diversity among the recovered samples, especially when using small masking ratios. This lack of diversity negatively impacts the effective recovery of source-like samples. Introducing the anchor (w/o Additional Obj) improves performance by providing guidance; however, excluding the additional objective prevents optimal results. Without this objective, directly pushing all recovered samples toward the anchor risks convergence to a collapsed solution, reducing the overall effectiveness of the recovery process.
>
> The detailed ablation study will be provided in the camera-ready version.
>
>
> | Variants           | HAR          | SSC          | MFD          |
> |--------------------|--------------|--------------|--------------|
> | w/o Early          | 90.24%±1.00% | 64.18%±0.58% | 92.91%±2.28% |
> | w/o Late           | 90.10%±1.06% | 64.34%±0.13% | 93.07%±1.15% |
> | w/o Recover        | 90.57%±0.91% | 64.50%±0.04% | 93.10%±0.14% |
> | w/o Complete       | 90.04%±1.13% | 63.96%±0.89% | 92.87%±2.27% |
> | w/o L_SegSim       | 90.17%±1.00% | 64.34%±0.32% | 93.19%±2.39% |
> | w/o Anchor         | 86.92%±2.95% | 63.57%±0.95% | 89.24%±5.58% |
> | w/o Additional Obj | 89.50%±0.57% | 64.17%±1.11% | 92.42%±5.08% |
> | TemSR              | 90.93%±0.54% | 64.72%±0.19% | 93.34%±2.31% |

---

> > ### Author Response · Authors · 2024-11-21
> > **Response [2/2]**
> >
> > **W2.** More explanation for the two key requirements
> >
> > **AWS:**
> >
> > 1. If the initial distribution significantly diverges from the source, aligning the two distributions becomes far more challenging. Initializing with a highly random or distant distribution forces the recovery model to search through a much larger solution space to approximate the source-like distribution. This expanded search space not only increases the complexity of finding an optimal solution but also heightens the risk of converging on a sub-optimal solution. A closer initial alignment with the source distribution narrows the search space, allowing the model to more efficiently and accurately capture essential source characteristics, leading to a more effective source-like domain.
> >
> > 2. If the initialized time points are randomly selected, they lack the sequential nature inherent in time-series data. Time-series models rely on the continuity of data points, as each point is not only a data value but also a part of a temporal sequence that defines dependencies across time. Randomized points disrupt this sequence, making it difficult for the recovery process to infer meaningful temporal patterns or to learn continuity in the data. This lack of order hinders the model’s ability to recover accurate temporal dependencies. By using continuous time points, we enable the recovery model to capture and recover the source temporal structure more effectively.
> >
> > We also conducted experiments using random initialization for the source-like domain. The results show that this random initialization significantly reduces performance. Furthermore, the standard deviation increases, highlighting the greater difficulty in identifying an optimal solution.
> >
> > | Variants                              | HAR          | SSC          | MFD          |
> > |---------------------------------------|--------------|--------------|--------------|
> > | w/ random initialization for src-like | 90.57%±2.25% | 63.82%±1.49% | 90.94%±7.68% |
> > | TemSR                                 | 90.93%±0.54% | 64.72%±0.19% | 93.34%±2.31% |
> >
> >
> > **Q1.** why minimizing the domain discrepancy between the source-like and target domains?
> >
> > **AWS:** As direct access to the source domain is not possible, we aim to create a source-like domain that closely approximates the distribution of the source domain. By minimizing the domain discrepancy between the source-like and target domains, we effectively reduce the discrepancy between the original source and target domains, facilitating successful domain adaptation. To enhance clarity, we have added relevant explanations in the problem formulation section.

---

> > > ### Author Response · Authors · 2024-11-25
> > > **Response Follow-up**
> > >
> > > Dear Reviewer gLQz,
> > >
> > > Thank you once again for taking the time to review our paper and provide valuable suggestions and comments. We would like to confirm whether the points addressed in our rebuttal are clear and aligned with your feedback. Please do not hesitate to let us know if any additional clarifications or details are needed.
> > >
> > > We greatly appreciate your time and consideration.

---

> > > ### Comment · Reviewer_gLQz · 2024-11-26
> > >
> > > Thank you for the feedback. Regarding Q1, it is still unclear to me what "the discrepancy between the original source and target domains" specifically refers to and why reducing it aligns with the objective of domain adaptation.

---

> > > > ### Author Response · Authors · 2024-11-28
> > > >
> > > > Thank you for your feedback. We appreciate the opportunity to address your concerns.
> > > >
> > > > Unsupervised Domain Adaptation (UDA) aims to bridge the gap between source and target domains to enable knowledge transfer from the labeled source domain to the unlabeled target domain. However, in the source-free UDA setting, the source domain is inaccessible during the adaptation stage, making it more challenging to directly address the original domain gap.
> > > >
> > > > To overcome this, we propose generating an effective source-like domain by recovering the temporal dependencies within the source domain. Since the source-like domain shares a similar distribution with the original source domain, minimizing the discrepancy between the source-like and target domains indirectly reduces the original source-target discrepancy. This approach facilitates successful domain adaptation despite the absence of the source domain during adaptation.
> > > >
> > > > Please feel free to share any further concerns or questions.

---

> > > > > ### Author Response · Authors · 2024-12-02
> > > > >
> > > > > Dear Reviewer gLQz,
> > > > >
> > > > > We hope our response has clarified your concerns.
> > > > >
> > > > > Please let us know if anything remains unclear or requires further explanation. We’d greatly appreciate your thoughts.

---

### Meta-Review · Area_Chair_E8cL · 2024-12-15

**Metareview:**

The paper introduces a method for source distribution recovery with a detailed methodological presentation and clearly defined variants. The idea of source distribution recovery is well-motivated, and the method section is clearly and comprehensively presented. The ablation study lacks thoroughness, as key designs mentioned in the method section, such as the consistent segment entropy loss, are not evaluated.

One reviewer accepted the paper, while two recommended rejection. After reviewing both the paper and the rebuttal, the AC decided that the paper should be rejected.

**Additional Comments On Reviewer Discussion:**

The reviewers expresses appreciation for the detailed response but remains unconvinced, specifically pointing out that temporal and spatial divergences in time-series data are not sufficiently addressed. The reviewers acknowledge that some works focus solely on temporal divergence but notes that many also consider both. Additionally, the reviewer is unclear about the meaning of "the discrepancy between the original source and target domains" and its relevance to the goal of domain adaptation.

---

### Decision · Program_Chairs · 2025-01-22

Reject